# Regulation of different phases of AMPA receptor intracellular transport by 4.1N and SAP97

Caroline Bonnet[1], Justine Charpentier[1], Natacha Retailleau[1], Daniel Choquet[1,2], Françoise Coussen[1]*

[1]University of Bordeaux, CNRS, Interdisciplinary Institute for Neuroscience, Bordeaux, France; [2]Bordeaux Imaging Center, Bordeaux, France

**Abstract** Changes in the number of synaptic AMPA receptors underlie many forms of synaptic plasticity. These variations are controlled by an interplay between their intracellular transport (IT), export to the plasma membrane (PM), stabilization at synapses, and recycling. The cytosolic C-terminal domain of the AMPAR GluA1 subunit is specifically associated with 4.1 N and SAP97. We analyze how interactions between GluA1 and 4.1N or SAP97 regulate IT and exocytosis in basal conditions and after cLTP induction. The down-regulation of 4.1N or SAP97 decreases GluA1 IT properties and export to the PM. The total deletion of its C-terminal fully suppresses its IT. Our results demonstrate that during basal transmission, the binding of 4.1N to GluA1 allows their exocytosis whereas the interaction with SAP97 is essential for GluA1 IT. During cLTP, the interaction of 4.1N with GluA1 allows its IT and exocytosis. Our results identify the differential roles of 4.1N and SAP97 in the control of various phases of GluA1 IT.

## Editor's evaluation

This important study by Bonnet et al. addresses the question of how AMPA receptor numbers at the synapse are regulated during basal conditions and during chemically induced Long Term Potentiation. Specifically, the study aims to determine the molecular mechanisms that contribute to the intracellular trafficking of AMPA receptors and determine their insertion into the synaptic plasma membrane. Using compelling methodology, the authors dissect the distinct roles of two proteins that bind to the C-terminal domain of the AMPA receptor subunit GluA1: 4.1N and SAP97. The findings will be of interest to anyone trying to understand molecular events contributing to synaptic plasticity in health and disease, and more broadly, the method could be adapted for tracking intracellular movements of a wide range of proteins.

*For correspondence:
fcoussen@u-bordeaux.fr

**Competing interest:** The authors declare that no competing interests exist.

## Introduction

AMPA-type glutamate receptors (AMPAR) are ionotropic tetrameric receptors activated by glutamate, the main excitatory neurotransmitter of the central nervous system. Their synaptic targeting, clustering, and immobilization in the post-synaptic density in front of glutamate release sites are crucial for efficient excitatory synaptic transmission. The number of neurotransmitter receptors, and particularly AMPAR, present at the synapse is regulated by a complex set of interdependent mechanisms going from biogenesis, IT (*Díaz-Alonso and Nicoll, 2021*; *Hiester et al., 2018*), externalization at the PM, lateral diffusion (*Choquet and Triller, 2013*), stabilization at synapses and trafficking in and out synaptic sites (*Groc and Choquet, 2020*). In highly polarized and spatially extended neurons, IT is of fundamental importance to distribute cargo over hundreds of micrometers and is likely finely tuned

and balanced to control receptor distribution. Accordingly, the IT of neosynthesized AMPAR plays a crucial role to transport them down the dendrites from the Golgi apparatus or Golgi outposts where they are matured after synthesis in the ER. After being released from the Golgi, the secretory vesicles containing the newly-synthesized AMPAR are trafficked to the PM through interaction with adaptor proteins and molecular motors to be finally exocytosed at the PM (*Schwenk et al., 2019*). We and others provided evidence that AMPAR IT is highly modulated by neuronal activity and this suggests that regulation of IT might be a core constituent of the control of synaptic strength during various forms of synaptic plasticity in neurons (*Hangen et al., 2018*; *Hoerndli et al., 2015*). However, despite its potential key role in synaptic regulation, and probable involvement in synaptopathies such as Huntington or Alzheimer diseases (*Mandal et al., 2011*), the molecular mechanisms that are involved in the regulation of AMPAR IT nevertheless remain largely unknown.

AMPAR IT is difficult to study in vertebrates due to the lack of reliable labeling methods and current limitations of imaging systems for detecting fast-moving, low-contrast small vesicles. In cultured rat hippocampal neurons, we have overcome these two hurdles using (1) a molecular tool allowing the retention and on-demand release of the newly synthesized AMPAR from the ER/Golgi and (2) the photobleaching of a portion of a dendrite followed by fast video acquisition (*Hangen et al., 2018*; *Rivera et al., 2000*). This allowed the characterization of IT of GluA1 AMPAR subunit, which forms homomeric calcium-permeable receptors, and can be inserted at synapses during synaptic plasticity (*Plant et al., 2006*; *Sanderson et al., 2012*).

We found that during chemically induced Long Term Potentiation of synaptic transmission (cLTP), the number and velocities of GluA1-containing vesicles are increased compared to the basal state (*Hangen et al., 2018*). These changes in vesicle velocities may be due to the diversity of molecular motors associated with AMPAR, although the exact motors involved are unknown. Molecular motors associate with their cargo through intermediate components, such as adaptors, scaffolds, and transmembrane proteins (*Klopfenstein et al., 2000*). AMPAR is part of a macromolecular complex composed of the receptor per se surrounded by a set of associated auxiliary (*Bissen et al., 2019*) and cytosolic proteins. Some of these intracellular partners have been shown to be associated with motor proteins and can modulate AMPAR surface expression.

Of particular interest in this regard, 4.1 N and SAP97 intracellular proteins are directly and specifically associated with GluA1 C-terminal (C-ter.) domain, the most variable domain between the different AMPAR subunits (*Diering and Huganir, 2018*; *Sans et al., 2001*; *Shen et al., 2000*). The C-ter. domain of GluA1 is particularly interesting for the regulation of IT as mutations on this domain modulate its transport and could be responsible for its upregulation during cLTP (*Hangen et al., 2018*). However, in the recent years, the C-ter. domain has been under intense scrutiny and its role in mediating synaptic plasticity has been debated. On the one hand, the C-ter. domain of native GluA1 and GluA2 has been suggested to be necessary and sufficient to drive NMDA receptor-dependent LTP and LTD, respectively (*Zhou et al., 2018*). On the other hand, the expression of heteromeric receptors containing the GluA1 subunit lacks the C-ter. domain maintains a normal basal trafficking and LTP at CA1 synapses in acute hippocampal slices (*Díaz-Alonso et al., 2020*). The method we developed (*Hangen et al., 2018*) and the results reported here will fuel this debate and allow to determine the exact contribution of GluA1 C-ter. domain for the regulation of IT properties of newly synthesized GluA1 subunit.

In red blood cells, the protein 4.1 (4.1R) is critical for the organization and maintenance of the spectrin–actin cytoskeleton and for the attachment of the cytoskeleton to the cell membrane. 4.1 N, the neuronal form of 4.1, may function to confer stability and plasticity to the neuronal membrane via interactions with multiple binding partners such as spectrin-actin–based cytoskeleton, integral membrane channels, and receptors. In neurons, 4.1 N associates specifically with GluA1 and colocalizes with AMPAR at excitatory synapses (*Walensky et al., 1999*). The C-ter. domain of 4.1 N mediates the interaction with the membrane-proximal region of GluA1. It has been suggested that 4.1 N regulates AMPAR trafficking by providing a critical link between the actin cytoskeleton and AMPAR (*Shen et al., 2000*). Phosphorylation of $S_{816}$ and $S_{818}$ residues in GluA1 regulates activity-dependent GluA1 insertion at the PM by enhancing the interaction between 4.1 N and GluA1. This suggests that 4.1 N is important for the expression of LTP, but doesn't affect basal synaptic transmission (*Lin et al., 2009*). However, while the regulation of GluA1 exocytosis by binding to 4.1 N has been established, its potential involvement in AMPAR IT still remains unknown.

SAP97, another important GluA1 C-ter. domain interactor is a member of the MAGUK family of proteins that play a major role in the trafficking and targeting of membrane ion channels and cytosolic structural proteins in multiple cell types (*Fourie et al., 2014*). Within neurons, SAP97 is localized throughout the secretory trafficking pathway and at the postsynaptic density (PSD). The role of SAP97 in the control of synaptic function is still unclear despite the fact that the PDZ2 domain of SAP97 interacts directly with the last four amino acids of GluA1 (*Cai et al., 2002*). The interaction between SAP97 and GluA1 occurs early in the secretory pathway, while the receptors are in the ER or cis-Golgi, and participates in its forward trafficking from the Golgi to the PM, suggesting that SAP97 acts on GluA1 solely before its synaptic insertion and that it does not play a major role in anchoring AMPAR at synapses (*Sans et al., 2001*; *Fourie et al., 2014*). SAP97 is a protein known for its involvement in NMDAR (*Jeyifous et al., 2009*) and AMPAR IT, thanks to its role as an adaptor protein between GluA1 and the actin-based motor MyoVI (*Wu et al., 2002*). However, the role of SAP97 in the trafficking and synaptic localization of AMPAR is still debated with conflicting results have been reported (*Fourie et al., 2014*; *Kay et al., 2022*; *Zhou et al., 2008*). Moreover, its role in the induction and maintenance of LTP is yet not well characterized.

Here, we have a unique experimental pipeline that allows us to differentiate IT from exocytosis of a given protein by measuring them independently. We report the role of the interactions between 4.1 N and SAP97 with the C-ter. domain of GluA1 by analyzing IT and exocytosis of newly synthesized GluA1 deleted for this domain under basal conditions and during synaptic activity. We identify different roles of the interactions between 4.1N-GluA1 and SAP97-GluA1 during basal transmission and after induction of cLTP in hippocampal cultured neurons.

## Results

To study the properties of AMPAR IT, we used cDNA constructs to express GluA1 and its different mutants subcloned in the ARIAD system and tagged at its N-terminus (ARIAD-Tag-GluA1) (*Hangen et al., 2018*). With this technology, receptors are retained in the ER in the basal state thanks to a conditional aggregation domain. Receptor release from the ER and follow-up secretion is tightly controlled with a cell-permeant drug (D/D solubilizer or ARIAD ligand: AL) that disrupts aggregation (*Rivera et al., 2000*). The synchronized release of receptors triggered by the addition of AL allows expressed proteins to progress through the secretory pathway in a synchronous manner, particularly adapted to monitor IT. Important features of this system include (1) no or low basal secretion and (2) a rapid and high level of secretion in response to the addition of AL (*Hangen et al., 2018*). This allowed us to measure three main parameters of AMPAR intracellular trafficking. First, the total number of GluA1-containing vesicles after the synchronized release of receptors was used as a measure of ER/Golgi export efficiency. Second, GluA1 vesicle transport properties (speed, fraction of time spent moving or pausing) were measured. Third, we determined the kinetic and extent of GluA1 appearance on the cell surface by live immunolabeling at various times after release. The comparative measurement of these different parameters allowed us to decipher finely the regulatory steps of GluA1 intracellular transport.

4.1 N and SAP97 are important proteins implicated in the regulation of AMPAR PM localization. Among all AMPAR subunits, these two proteins are specifically associated with the GluA1 C-ter. domain (*Lin et al., 2009*; *Rouach et al., 2005*; *Schwenk et al., 2014*), (*Figure 1—figure supplement 1A*). We analyzed how interactions between these associated proteins and GluA1 regulate AMPAR IT in basal conditions and after induction of cLTP in cultured rat hippocampal neurons.

### GluA1 intracellular transport and exocytosis are dependent on the expression of 4.1N or SAP97

4.1 N and SAP97 participate in the biosynthesis and processing of AMPAR in the hippocampus (*Sans et al., 2001*; *Shen et al., 2000*). Previous studies established that knocking down 4.1 N by expression of a specific sh-RNA substantially reduced the frequency of GluA1 exocytosis, indicating that 4.1 N is critical for GluA1 insertion at the PM (*Lin et al., 2009*). On the other hand, SAP97 has been shown to associate with GluA1 containing AMPAR while they are in the ER, with SAP97 dissociating from the receptor at the PM (*Sans et al., 2001*).

We decided to knock down each of these two proteins independently and analyze IT and externalization of newly synthesized GluA1 in basal condition. We expressed sh-RNAs against 4.1 N or SAP97 or their corresponding control (scramble) and analyzed the trafficking of ARIAD-Tag-GluA1 after the addition of the ligand to release the protein from the ER (*Figure 1*).

We first controlled the efficacy of the sh-RNA in rat-cultured hippocampal neurons by expressing viruses containing respectively scramble RNA (scr.), sh-RNA against 4.1 N (sh-4.1N) or against SAP97 (sh-SAP97) (*Figure 1A*). Expression of the endogenous proteins were significantly decreased by expression of the corresponding sh-RNA. To test the specificity of the sh-RNA, we expressed 4.1 N or SAP97 or the corresponding rescue proteins in COS-7 cells together with the scr.-RNA or the sh-RNA and quantified expression of 4.1 N and SAP97 by western blot analysis (*Figure 1B*). As in neurons, expression of sh-RNA decreased the expression of the corresponding wild type proteins without affecting the expression of the corresponding rescue proteins showing the specificity of our sh-RNA.

We then analyzed the parameters of GluA1 IT when 4.1 N or SAP97 sh-RNA or corresponding scr.-RNA were expressed (*Figure 1C–G*). We expressed GluA1 subcloned in the ARIAD vector, induced the transport of the protein by the addition of the AL, and analyzed the transport 30–60 min after the addition of the AL. During this time window, the vesicles are traveling in dendrites with almost no background signal coming from the PM (*Hangen et al., 2018*). In both cases, the total number of vesicles transporting GluA1 was decreased, although less drastically when 4.1 N was knocked down than when SAP97 was knocked down (*Figure 1C*).

For each cell, we traced the corresponding kymographs and calculated the mean speed of the vesicles (*Figure 1D–E*, Sup. *Figure 1B*). We found similar values for speed for the OUT (from the cell body to the dendrite) and for the IN (from the dendrite to the cell body) directions (*Figure 1—figure supplement 1C*). We thus decided to pool the speeds of transport of the OUT and the IN directions. The mean speed of the vesicles was only decreased by 13% by expression of the sh-4.1N compared to its scr.-RNA (*Figure 1E*, *Figure 1—figure supplement 1C*). When sh-SAP97 was expressed, the mean speed was decreased by 25% compared to the corresponding scr.-RNA. Thanks to the kymographs, we calculated the percentage of time spent in the moving and in the pausing states for each vesicle (*Figure 1F–G*, *Figure 1—figure supplement 1D–E*). The time spent moving was decreased by 8% when the sh-4.1N was expressed to the benefit of the pausing time. Indeed, when the expression of SAP97 was decreased, the time spent moving by a vesicle was decreased by 20% to the benefit of the time spent in pause.

We then analyzed the kinetics of externalization of GluA1 in the same conditions as for the IT experiments. We performed live extracellular labeling of GluA1 at 45 and 60 min after the addition of the AL on hippocampal rat cultured neurons (*Figure 1H–I*). Expression of sh-4.1N decreased massively the externalization of GluA1 compared to its externalization with the expression of the corresponding scr. (*Figure 1H*). The 4.1 N rescue protein could prevent this decrease in the rate of externalization when expressed together with the sh-4.1N. Expression of sh-SAP97 also decreased the rate of externalization of GluA1 to the same extent as sh-4.1N did (*Figure 1I*). Expression of SAP97 rescue protein partially restored the externalization of GluA1.

In conclusion, reducing the expression of 4.1 N and SAP97 both diminished GluA1 IT in rat-cultured hippocampal neurons. However, the effects were all less drastic when 4.1 N expression was decreased than when SAP97 was. This was the case for the decrease in the number of vesicles released upon the addition of AL, for the decrease in the vesicle speed, and for the increase in pausing time. However, we found that the externalization of GluA1 at the PM was equally inhibited by the absence of 4.1 N or SAP97. Because the down-regulation of 4.1 N or SAP97 could have indirect effects on GluA1 transport properties, we then studied the impact of GluA1 mutations that inhibit its interaction with these proteins.

## AMPAR traffic is regulated by the interaction between GluA1 C-ter. domain and 4.1N or SAP97

GluA1 IT and PM localization is dependent on the expression of 4.1 N and SAP97. Knocking down either 4.1 N or SAP97 decreases massively the exit of GluA1 from the ER-Golgi and impacts IT and exocytosis of the receptor to the PM. However, the overall impact of SAP97 is more drastic on IT whereas we found the same effect for both conditions on the PM localization of GluA1. This may be because the interaction between 4.1 N and GluA1 might be necessary mainly for the exocytosis of

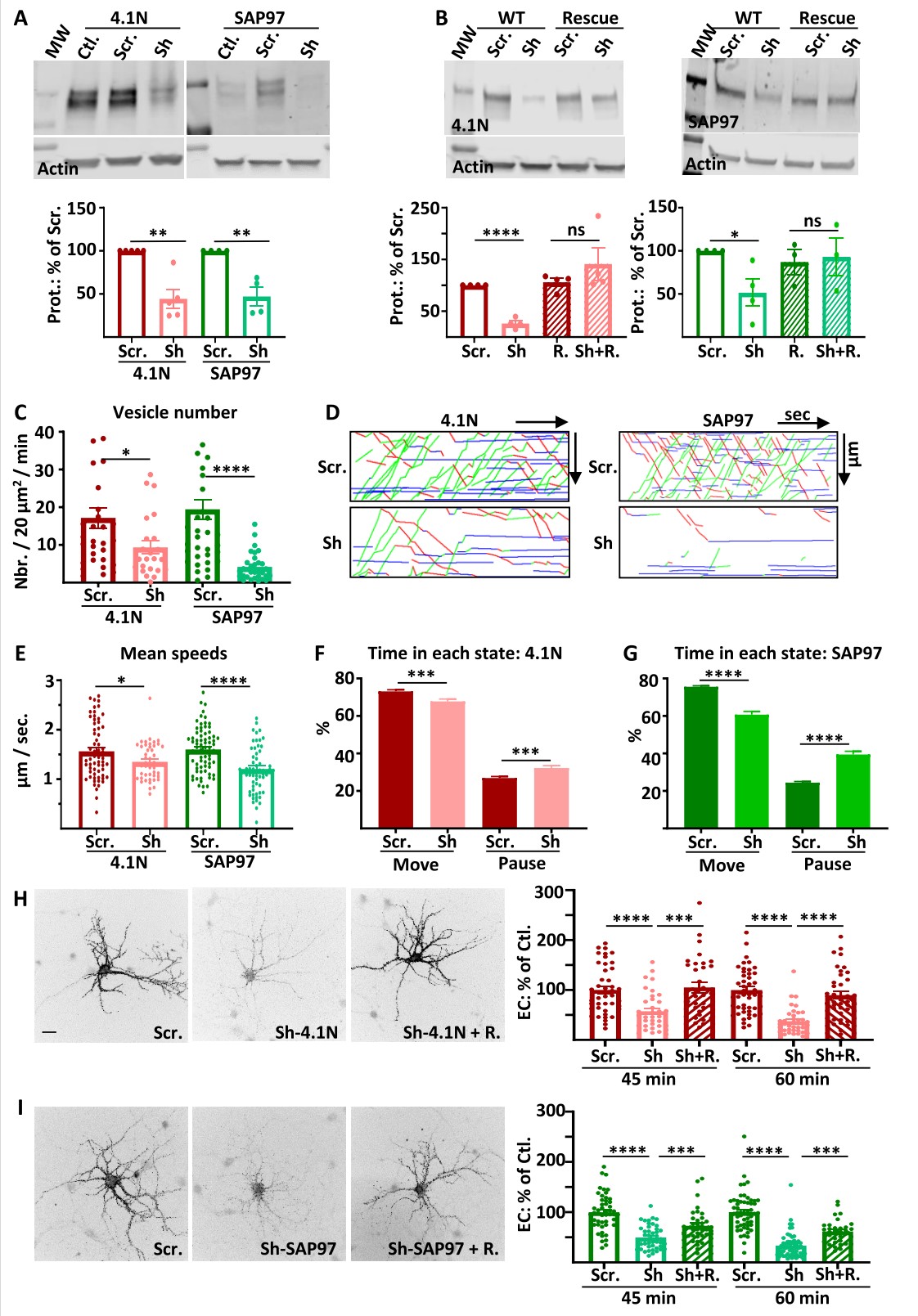

**Figure 1.** Intracellular transport and exocytosis of GluA1 are dependent on the expression of 4.1N and SAP97. ( **A**) Top: Western blots of 4.1N and SAP97 expression in cultured rat hippocampal neurons after virus infection with scramble-RNA (scr.) or sh-RNA against 4.1N and SAP97; bottom: quantification of proteins normalized with actin on the scr. condition (sh-4.1N: 44.4 +/−11.2%,n=5, sh-SAP97: 58.3 +/−7.3%, n=4). (**B**) Top: Western blots showing the expression of 4.1N and SAP97 WT and rescue after transfection of the proteins in COS-7 cells; bottom: quantifications normalized with

*Figure 1 continued on next page*

*Figure 1 continued*

actin on the scr. condition (sh-4.1N: 26.2 +/−10.1%, sh-SAP97: 47.1 +/−10.7%, n=4; for rescue proteins; scr.: 105.9 +/−7.9%, sh-4.1N: 141.2 +/−3.1%, scr.: 87.0 +/−1.5%, sh-SAP97: 93.0 +/−2.2%, n=4). (**C** to **G**) Parameters of intracellular transport of ARIAD-TdTom-GluA1 expressed with scramble-RNA (scr.) or sh-RNA against 4.1N and SAP97. (**C**) Vesicle number (vesicles/20 μm$^2$/min; scr. 4.1 N: 17.2 +/−2.7, sh-4.1N: 9.3 +/−1.7, n=3 scr. SAP97: 19.5 +/−2.5; sh-SAP97: 4.2 +/−0.7, n=4), (**D**) Representative kymographs of the routes of the vesicles in the function of the time in the video. (**E**) Mean speeds of the vesicles in control (expression of scr.) and when 4.1 N or SAP97 are decreased (expression of sh) (μm/s; scr. 4.1 N: 1.56 +/−0.07, sh-4.1N: 1.35 +/−0.05, n=3; scr. SAP97: 1.60 +/- 0.05, sh-SAP97: 1.21 +/- 0.07, n=4), (**F–G**) Time spent by a vesicle in a moving state (Move) or in pausing state (Pause) (% Move: scr. 4.1N: 73.13 +/−0.83, sh-4.1N: 67.77 +/−1.25; % pause: scr. 4.1N: 26.87 +/−0.83, sh-4.1N: 32.22 +/−1.25) and (% Move: scr. SAP97: 75.57 +/−5.89, sh-SAP97: 60.68 +/−1.74; % pause: scr. SAP97: 24.43 +/−5.89, sh-SAP97: 39.32 +/−1.75) (n=3) (**H**) Representative image of live extracellular labeling of ARIAD-GFP-GluA1 after 45 and 60 min. of incubation with AL expressed with sh-RNA for 4.1N with or without the corresponding rescue proteins and quantifications (% of cle. 45 min. after AL; sh-4.1N: 57.36 +/- 6.27, sh-4.1N on rescue: 105.00 +/- 10.34; 60 min after AL; sh-4.1N: 37.29 +/−4.16, sh-4.1N on rescue: 89.73 +/−7.81) (n=3). (**I**) Representative image of live extracellular labeling of ARIAD -GFP-GluA1 after 45 and 60 min of incubation with AL expressed with sh-RNA for SAP97 with or without the corresponding rescue proteins and quantifications (% of cle. 45 min after AL; sh-SAP97: 49.64 +/−3.41, sh-SAP97 on rescue: 72.91+/−5.58, 60; 60 min after AL; sh-SAP97: 33.33 +/−3.29, sh-SAP97 on rescue: 61.69 +/−4.32) (n=3). The 100% values for **H** and **I** correspond to the extracellular labeling of the control (Scramble: Scr.) for the same times of incubation with AL. Scale bar: 25 μm.

The online version of this article includes the following source data and figure supplement(s) for figure 1:

**Source data 1.** Individual data values for the bar graphs in panels A, B, C, E, F, H and I.

**Figure supplement 1.** Intracellular transport and exocytosis of GluA1 are regulated by 4.1N and SAP97.

**Figure supplement 1—source data 1.** Individual data values for the bar graphs in panels C and D.

the receptor at the PM. We thus decided to analyze if the interaction between GluA1 and 4.1 N or GluA1 and SAP97 is important for the intracellular transport and exocytosis of the newly synthesized receptor in basal synaptic transmission (*Figure 2*).

We first checked by co-immunoprecipitation experiments if endogenous GluA1 and 4.1 N and GluA1 and SAP97 are interacting in our model of rat-cultured hippocampal neurons (*Figure 2A*). Indeed, we found that immunoprecipitation of 4.1 N or SAP97 co-immunoprecipitated GluA1 and immunoprecipitation of GluA1 co-immunoprecipitated 4.1 N and SAP97. It has been shown that GluA1 binds SAP97 by its last four amino acids (*Leonard et al., 1998*) whereas it binds 4.1 N on a peptide domain localized just after its fourth transmembrane domain (*Shen et al., 2000*). We designed different GluA1 mutants in order to study the impact of these interactions on GluA1 IT (*Figure 2B*). We deleted the entire C-ter. domain of GluA1 (deletion of the last 78 amino acids of GluA1 leaving only four amino acids after the last transmembrane domain, Δ78) or each of the interaction sites for 4.1 N (Δ4.1N: deletion of 14 amino acids and five amino acids after the last transmembrane domain) and for SAP97 (Δ7: deletion of the last seven amino acids of GluA1). For each mutant, we studied their PM localization as a function of the time of incubation with the AL and the characteristic of their IT.

The externalization of newly synthesized GFP-GluA1-Δ78 or GFP-GluA1-WT was monitored by live immunolabeling with an antibody directed against GFP (*Figure 2C and D*). Quantification of the GFP staining revealed an almost complete disappearance of GluA1-Δ78 externalizations. This experiment demonstrates that the C-ter. domain is necessary for newly synthesized GluA1 to be externalized at the PM, even 2 hr after triggering GluA1 ER exit.

We then studied if the interaction between GluA1 and 4.1 N or SAP97 is necessary for the localization of GluA1 at the PM (*Figure 2C and E*, *Figure 2—figure supplement 1A*). We performed extracellular labeling of GFP-GluA1-WT and the mutants, GFP-GluA1-Δ4.1N, and GFP-GluA1-Δ7 respectively deleted for their binding site for 4.1 N or SAP97, after induction of IT by addition of the AL during different times. For analysis of these experiments, we normalized the externalization values of the mutants to that of GFP-GluA1-WT at the corresponding times of incubation with AL in paired experiments. Both mutants were less exocytosed than GluA1-WT from 30 min to 2 hr after the addition of the AL. At each time point, the extracellular labeling of GluA1 lacking the 4.1 N binding site (Δ4.1N) was inferior to the one lacking the SAP97 binding site (Δ7). Indeed, when we quantified at 90 and 120 min. the difference in exocytosis between the two mutants, we found that this difference between Δ4.1N and Δ7 proteins was highly significant (*Figure 2—figure supplement 1A*). Exocytosis of the Δ4.1N mutant is less important than for the Δ7 mutant (*Supplementary file 1*). This result shows that the interaction of GluA1 with 4.1 N or SAP97 plays a role in the surface expression of newly synthesized GluA1.

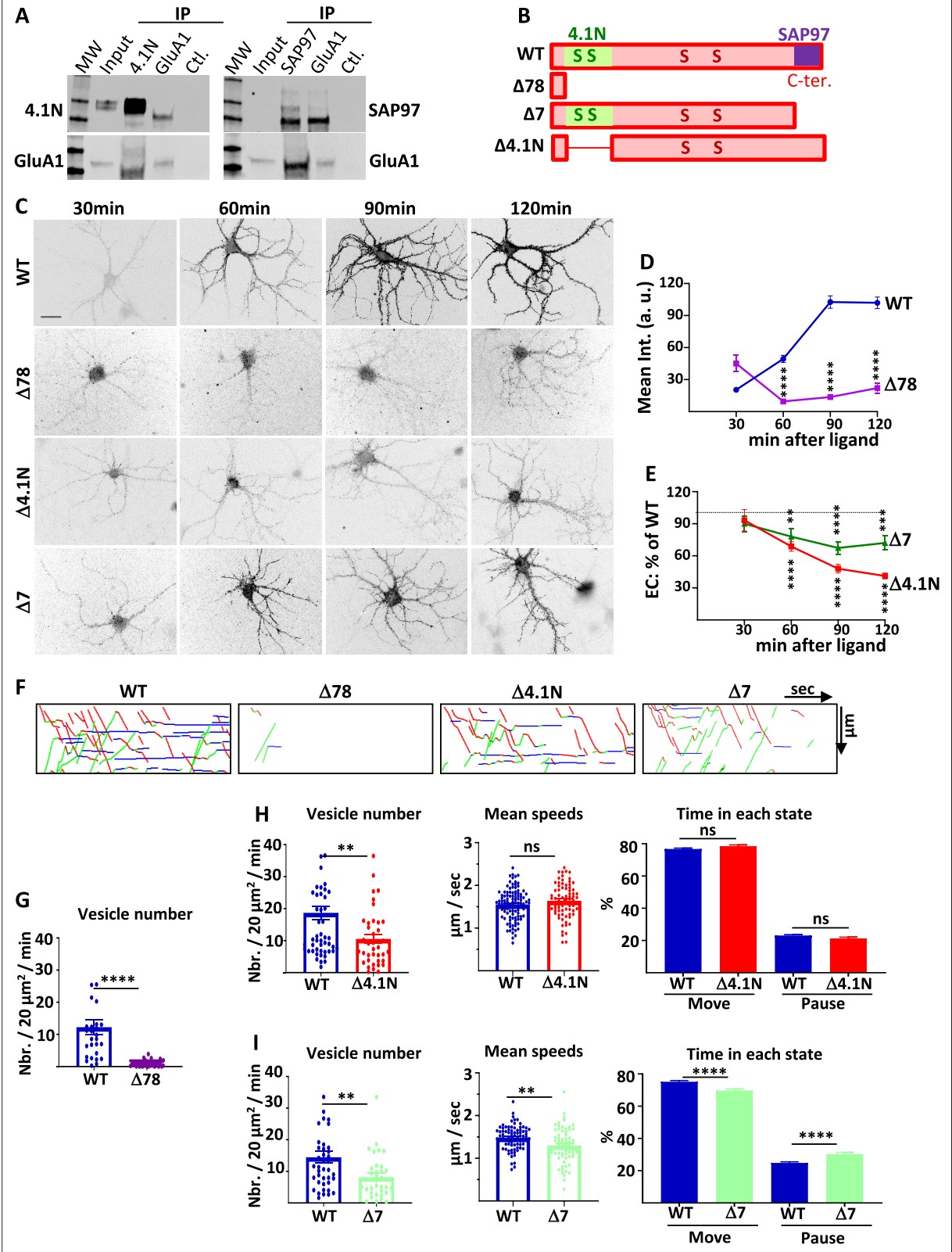

**Figure 2.** 4.1N/GluA1 and SAP97/GluA1 interactions differently regulate GluA1 traffic in basal transmission. (**A**) Co-immunoprecipitation of endogenous GluA1 with 4.1N and SAP97 in cultured rat hippocampal neurons. Control (Ctl.) is performed without an antibody. Western blot of GluA1, 4.1N and SAP97 as indicated. (**B**) Diagram of the different truncated mutants on the C-terminal (C-ter.) domain of GluA1. (**C**) Representative images of live labeling of ARIAD-GFP-GluA1 after the addition of AL during different times as indicated. Scale bar: 25 µm. (**D**) Quantification of the exit of ARIAD-GFP-GluA1-

*Figure 2 continued on next page*

*Figure 2 continued*

WT (WT) and ARIAD-GFP-GluA1-Δ78 (Δ78) over time after the addition of AL. For the WT, 100% of exit is taken after 120 min of addition of AL. (WT versus Δ78, arbitrary unit (a. u).): 30 min, 20.43 +/−2.14 vs 45.02 +/−7.75; 60 min, 49.39 +/−3.42 vs 9.54 +/−1.95; 90 min, 102.68 +/−5.83 vs 13.64 +/−1.70; 120 min, 101.95 +/−5.37 vs 22.02 +/−4.59) (E) Quantification of the exit of ARIAD-GFP-GluA1-Δ4.1N (Δ4.1N) and ARIAD-GFP-GluA1-Δ7 (Δ7) over time after addition of AL induction (Δ4.1N: 30 min, 93.22 +/−10.00; 60 min, 68.75 +/−4.43; 90 min, 47.94 +/−4.01; 120 min, 41.19 +/− 2.70; Δ7: 30 min, 90.07 +/−7.42; 60 min, 77.98 +/−7.26; 90 min, 67.35 +/−5.68; 120 min, 72.04 +/−6.60). The 100% values correspond to the value of the WT for the same time, shown by a dotted line. (F) Traced kymographs for the different mutants. (G) Number of vesicles detected for the ARIAD-TdTom-GluA1-WT (WT) and the ARIAD-TdTom-GluA1-Δ78 (Δ78) (vesicles / 20 µm2/min; GluA1-WT: 12 +/−2.3, Δ78: 0.86 +/−0.2, n=4). (H) Parameters of intracellular transport for ARIAD-TdTom-GluA1-Δ4.1N (Δ4.1). Vesicle number (vesicles/20 µm$^2$/min; GluA1-WT: 18.68 +/−2.05, Δ4.1N: 10.52 +/−1.40, n=5), mean speeds (µm/s; WT: 1.55 +/−0.03, Δ4.1N: 1.64 +/−0.05) and percentage of time in each state (% Move WT: 76.75 +/−0.53 %, Δ4.1N: 78.52 +/−0.78%; % pause: WT: 23.10 +/−0.53 %, Δ4.1N: 21.31 +/−0.78 %) (n=5). (I) Parameters of intracellular transport for the ARIAD-TdTom-GluA1-Δ7 (Δ7). Vesicle number (vesicles/20 µm$^2$/min; GluA1-WT: 14.45 +/−1.82, Δ7: 8.16 +/−1.21), mean speeds (µm/s; WT: 1.48+/−0.03, Δ7: 1.30+/−0.05) and percentage of time in each state (% Move: WT: 75.18+/−0.70%, Δ7: 69.67+/−1.04%; % pause: WT: 24.82+/−0.69%, Δ7: 30.33+/−1.04%) (n=4).

The online version of this article includes the following source data and figure supplement(s) for figure 2:

**Source data 1.** Individual data values for the bar graphs in panels D, E, G, H and I.

**Figure supplement 1.** 4.1 N/GluA1 and SAP97/GluA1 interactions differently regulate GluA1 traffic in basal transmission.

**Figure supplement 1—source data 1.** Individual data values for the bar graphs in panels A and C.

This lack of normal exocytosis of the mutants can be due either to an inhibition of externalization or to a decrease in their IT. We thus analyzed the IT parameters of the different mutants taking GluA1-WT as a control (*Figure 2F–I*, Sup. *Figure 2—figure supplement 1B and C*). For these experiments, we expressed ARIAD-TdTom-GluA1, WT, or mutants, in order to be in the best conditions to detect the transport vesicles. We first analyzed the IT of the Δ78 mutants (*Figure 2F–G*). The number of vesicles transporting this mutant was very low compared to GluA1-WT and this prevented the analysis of their IT parameters. The C-ter. domain of GluA1 is thus mandatory for the exit of newly synthesized GluA1 from the ER and the Golgi, likely explaining its requirement for GluA1 surface expression.

We then characterized the IT of the mutant deleted for the 4.1 N binding site (Δ4.1N) (*Lin et al., 2009*; *Figure 2F and H*, *Figure 2—figure supplement 1B and C*). The number of vesicles transporting the protein was decreased compared to GluA1-WT. This is in accordance with a decrease in the number of vesicles that we found when 4.1 N was knocked down by the expression of the sh-4.1N. In contrast, the mean speeds of the vesicles were the same for GluA1-WT and GluA1-Δ4.1N. The time spent in each state was similar for the two proteins. These results indicate that binding of GluA1 to 4.1 N is important for its ER or Golgi export and exocytosis at the PM but does not affect its IT once the vesicles are released from the Golgi apparatus.

We then analyzed the IT of the mutant deleted for the last seven amino acids corresponding to the binding site of SAP97 (Δ7) (*Zhou et al., 2008*; *Figure 2F1*, *Figure 2—figure supplement 1B and C*). For this mutant, the number of vesicles released was decreased compared to GluA1-WT, as for GluA1-Δ4.1N. We also found a highly significant effect of the Δ7 mutant deletions of the PDZ binding domain on all the GluA1 IT parameters. First, the mean speed was decreased by 12% for the Δ7 mutants. Moreover, the time spent in a moving state was decreased by 7% and, conversely, the percentage of time in pause was increased by 22% compared to the WT. All these changes, although relatively modest, are significant and can have important functional impacts when cumulated over time. This corresponds to what we found for IT properties when the expression of SAP97 was decreased: modified ER/Golgi export and PM exocytosis, reduced speed, and increased time in pause.

For these two mutants, the vesicle number is similarly decreased but if the mean speed is also decreased for the Δ7 this is not the case for the Δ4.1N which has a completely normal IT speed. On the contrary, the exocytosis is largely decreased for the Δ4.1N protein (*Supplementary file 1*).

## AMPAR traffic is regulated by the specific interaction between GluA1 C-ter. domain with 4.1N

Since we observed an impact on GluA1 IT when 4.1 N was knocked down, we were surprised by the absence of impact on IT of the Δ4.1N deletion on GluA1 IT. We thus decided to analyze the characteristics of IT with a GluA1 mutant that does not bind 4.1 N and has the same C-ter. domain length (*Figure 3*, *Figure 3—figure supplement 1*). During LTP, protein kinase C (PKC) phosphorylates the serine 816 ($S_{816}$) and serine 818 ($S_{818}$) residues in the GluA1 C-ter. domain. These phosphorylations

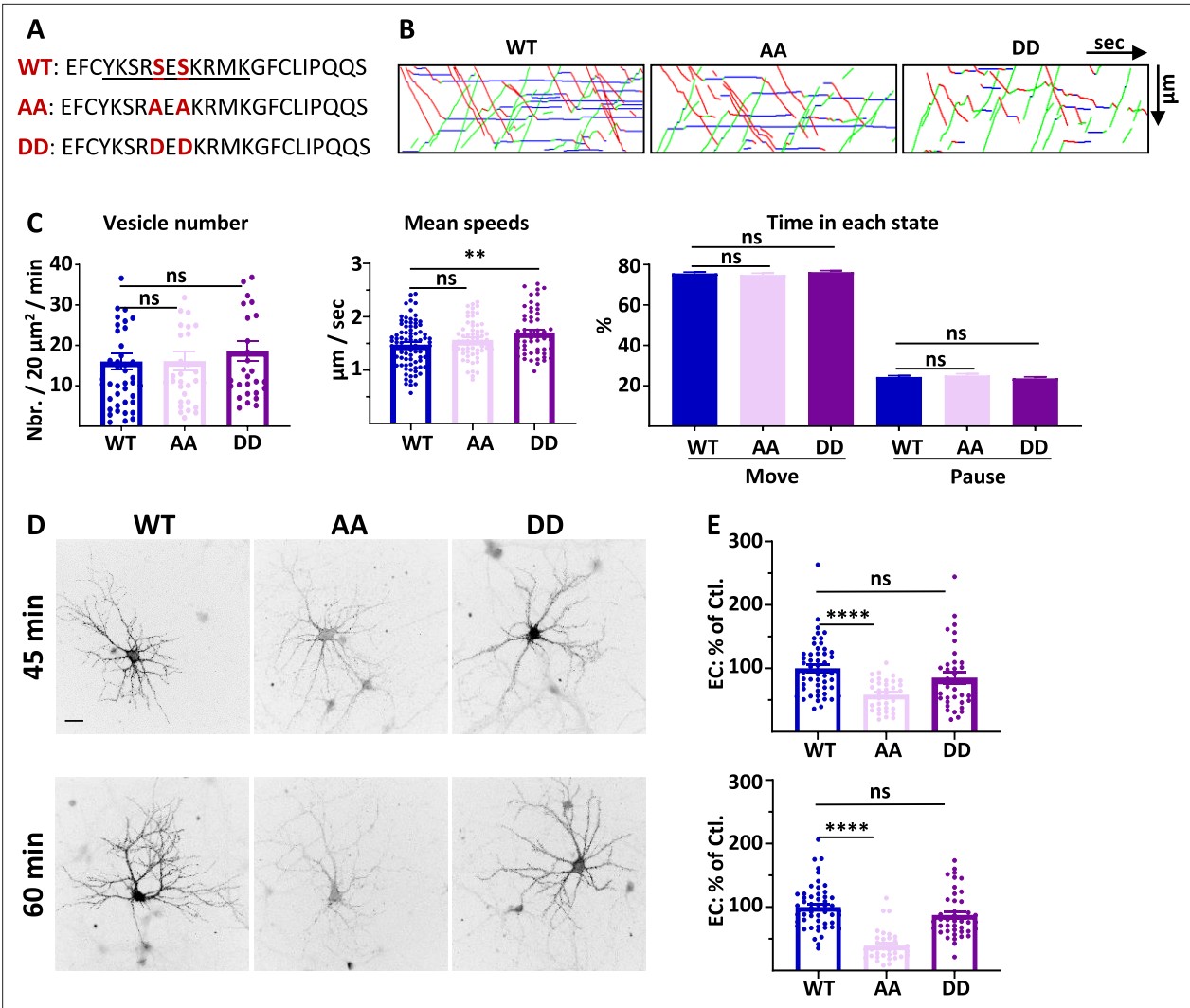

**Figure 3.** 4.1 N/GluA1 interaction is only necessary for the exocytosis of GluA1 in basal conditions. (**A**) Amino acid sequences representing the binding site of 4.1N on the C-ter. domain of GluA1 and mutations corresponding to the $S_{816}A$ $S_{818}A$ (AA) and $S_{816}D$ $S_{818}D$ (DD) mutants. (**B**) Representative traced kymographs for ARIAD-TdTom-GluA1-WT (WT), ARIAD-TdTom-GluA1-AA (AA), and ARIAD-TdTom-GluA1-DD (DD). (**C**) Parameters of IT for the WT, AA, and DD constructs. vesicle number (vesicles/20 µm²/min; GluA1-WT: 16.01 +/−2.05, AA: 16.10 +/−2.34, DD: 18.61 +/−2.56), mean speeds (µm/s; WT: 1.48 +/−0.04, AA: 1.57 +/−0.04, DD: 1.71 +/−0.05), percentage of time in each state (Move: WT: 75.64 +/−0.69%, AA: 74.90 +/−0.84%, DD: 76.31 +/−0.75%; Pause: WT: 24.36 +/−0.69%, AA: 25.10 +/−0.84%, DD: 23.70 +/−0.75%) (n=6). (**D**) Representative images of live labeling of ARIAD-GFP-GluA1-WT (WT), ARIAD-GFP-GluA1-AA (AA), and ARIAD-GFP-GluA1-DD (DD) after the addition of AL during different time as indicated. Scale bar: 25 µm. (**E**) Quantification of live labeling after 45 and 60 min of incubation with AL normalized on the WT after the same time of incubation with AL (% of EC labeling at 45min WT: 100 +/−6.0; AA: 58.44 +/−3.99; DD: 85.39 +/−8.75; at 60 min WT: 100 +/−4.76; AA: 39.27 +/−4.01; DD: 85.57 +/−5.39) (n=3). The 100% values correspond to the extracellular labeling of the WT (Ctl.) for the same times of incubation with AL.

The online version of this article includes the following source data and figure supplement(s) for figure 3:

**Source data 1.** Individual data values for the bar graphs in panels D, E, G, H and I.

**Figure supplement 1.** 4.1 N/GluA1 interaction is only necessary for the exocytosis of GluA1 in basal conditions.

**Figure supplement 1—source data 1.** Individual data values for the bar graphs in panels B and D.

enhance 4.1 N binding to GluA1 and facilitate GluA1 insertion at the PM. When these two serines are replaced by alanines, the interaction with 4.1 N is abolished (*Lin et al., 2009*). We constructed the corresponding $S_{816}A-S_{818}A$ (AA) or the LTP-like constitutively phosphorylated $S_{816}D-S_{818}D$ (DD) mutants of ARIAD-TdTom-GluA1 and ARIAD-GFP-GluA1 (*Figure 3A*). These constructs allowed us to study with more specificity the impact of the interaction between 4.1 N and GluA1.

We first performed IT experiments with these GluA1 mutants (*Figure 3B and C*; *Figure 3—figure supplement 1*). The number of vesicles was similar for the WT AA and DD mutants (*Figure 3B and C*; *Figure 3—figure supplement 1A*). We then analyzed the mean speed of the vesicles containing each protein (*Figure 3C*, *Figure 3—figure supplement 1B and C*). As for the Δ4.1N mutant, we did not find any difference in the mean IT speed between WT and AA proteins. However, the mean speed for the DD mutant was significantly increased compared to the WT, specifically for the OUT direction (Sup. *Figure 3B*). The frequency distribution of the speeds was identical for the three conditions in both OUT and IN directions (Sup. *Figure 3C*). The percentage of time spent in moving and pausing states was also identical for the three proteins WT, AA, and DD (*Figure 3D*, *Figure 3—figure supplement 1D*). This shows that, in the basal state, GluA1 IT is not dependent on the binding of 4.1 N since the AA mutant has the same properties than the WT. This result is in accordance with the observation reported above that the GluA1-WT and GluA1-Δ4.1N IT are similar. We thus concluded that the impact of knocking down 4.1 N on GluA1 IT, in basal condition, does not originate from an interaction between GluA1 and 4.1 N.

We then measured the export to the PM of GluA1 WT and the corresponding mutants 45 min and 60 min after the addition of AL (*Figure 3D and E*). Already after 45 min of IT induction, we found that the exit of GluA1 was strongly decreased for the GluA1-AA mutant but comparable with the WT for the DD mutant. This difference in exocytosis of GluA1 was accentuated when the neurons were incubated for 60 min with the AL.

In conclusion, during basal transmission, the interaction between GluA1 and 4.1 N has a fundamental role in the exocytosis at the PM of GluA1 without regulating its IT (*Supplementary file 1*).

## SAP97 regulates GluA1 traffic during cLTP

We have shown previously that GluA1 IT is strongly increased 25–40 min after induction of cLTP in cultured rat hippocampal neurons (*Hangen et al., 2018*). In this condition, the number of vesicles is increased by 140%, the speed is higher and the pausing time is lower. Here, we applied the same cLTP protocol to investigate the role of SAP97 (and then 4.1 N below) in the regulation of the late phase of cLTP (200 μM Glycine, 20 μM bicuculline, 0 mM Mg ++ during 3 min. and then return to normal media). Videos were acquired 25–40 min after induction of cLTP. For some experiments, a video was acquired before induction for 1–2 cells expressing GluA1-WT control to verify the efficacy of our cLTP protocol (Sup. *Figure 4A*). We analyzed the ER/Golgi export, transport, and PM export properties of GluA1 after cLTP in the function of both the level of expression of SAP97 and the binding of GluA1 to SAP97 by expressing scramble or sh-SAP97 and comparing IT between GluA1-WT and GluA1-Δ7 mutant (*Figure 4*, *Figure 4—figure supplement 1*).

The number of vesicles containing GluA1 after cLTP was strongly decreased by expression of sh-SAP97 as compared to cLTP in control conditions and when we expressed GluA1-Δ7 compared to GluA1-WT (*Figure 4A and B*, *Figure 4—figure supplement 1B*). These effects were identical to those observed in basal conditions both for sh-SAP97 and Δ7 mutant conditions (% decrease of vesicles: control/sh-SAP97 in basal: 78.4 %, after cLTP: 67.1%; control/Δ7 in basal: 27.2%, after cLTP: 33.8 %). Similarly to the effects observed in basal conditions, the mean speeds were also decreased after cLTP in the sh-SAP97 condition and when GluA1-Δ7 truncation mutant was expressed (*Figure 4B*, *Figure 4—figure supplement 1C*). The time spent in the moving state was also similarly decreased compared to the corresponding controls (*Figure 4B*). These effects are in accordance with the impact that we already found during basal transmission.

We next analyzed the impact of the GluA1-Δ7 truncation on its externalization to the PM after induction of cLTP (*Figure 4C and D*). We induced cLTP 20 min after the addition of AL and performed live immunolabeling of neosynthesized GluA1 25 min (total with AL 45 min) or 40 min (total with AL 60 min) after induction of cLTP. cLTP significantly increased the exit of GluA1-WT both at 45 and 60 min but had no impact on the externalization of GluA1-Δ7, neither at 45 min nor at 60 min. Altogether, this shows that downregulating SAP97 or preventing its binding to GluA1 abolished the cLTP-induced regulation of GluA1 ER/Golgi exit, IT, and PM externalization (*Supplementary file 1*).

## 4.1N regulates GluA1 traffic during cLTP

Disrupting 4.1N-dependent GluA1 PM insertion decreases the surface expression of GluA1 and the expression of long-term potentiation showing that 4.1 N is required for activity-dependent GluA1

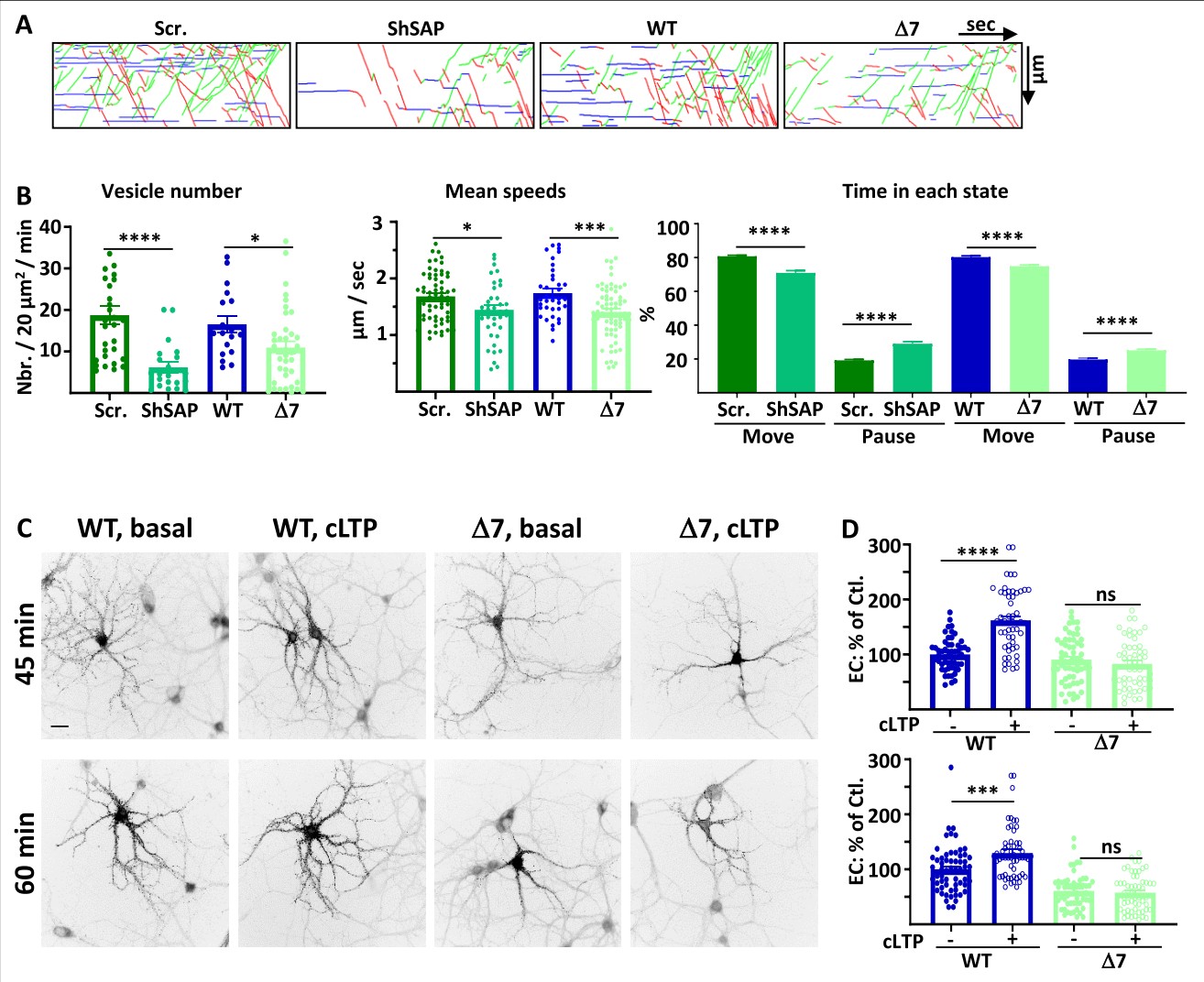

**Figure 4.** SAP97 and SAP97/GluA1 interaction regulate GluA1 trafficking during cLTP. (**A**) Representative traced kymographs after LTP for ARIAD-TdTom-GluA1-WT co-transfected with scramble-RNA (Scr.) or with sh-SAP97 (ShSAP), ARIAD-TdTom-GluA1-WT (WT) and ARIAD-TdTom-GluA1-Δ7 (Δ7). (**B**) IT parameters of GluA1 in the different conditions as indicated 25-40 min after induction of cLTP: Vesicle number (vesicles/20 µm²/min; scr.: 18.77 +/−2.29, Sh-SAP97: 6.18 +/−1.25; WT: 16.55 +/−1.92, Δ7: 10.95 +/−1.50), mean speeds (µm/s; scr.: 1.68 +/−0.06, Sh-SAP97: 1.45 +/−0.09, WT: 1.74 +/−0.08; Δ7: 1.41 +/−0.06), time in each state (Move: scr.: 80.82 +/− 0.53%, Sh-SAP97: 70.99 +/−1.29%; pause: scr.: 19.18 +/−0.53%, Sh-SAP97: 29.00 +/−1.30%, and Move: WT: 79.63 +/−0.75%, Δ7: 76.45 +/−0.73%; pause: WT: 20.37 +/−0.75%, Δ7: 23.55 +/− 0.73%) (n=3). (**C**) Representative images of neurons for each condition 45-60 min after the addition of the AL, 25-40 after induction of cLTP. Scale bar: 25 µm. (**D**) Quantification of live immunolabeling before and after induction of cLTP for the GluA1-WT (WT) and GluA1-Δ7 (Δ7) mutant (% PM localization at 45 min: WT before LTP:100 +/−3.89, after LTP: 162.24 +/−7.71, Δ7 before LTP: 90.71 +/−5.50, after LTP: 82.78 +/−6.09; % PM localization at 60 min: WT before LTP: 100 +/−5.63, after LTP: 129.63 +/−6.22, Δ7 before LTP: 60.82 +/−4.27, after LTP: 57.62 +/−4.50) (n=3). Control (Ctl.) corresponds to extracellular (EC) labeling of the WT before LTP normalized to 100%.

The online version of this article includes the following source data and figure supplement(s) for figure 4:

**Source data 1.** Individual data values for the bar graphs in panels B and D.

**Figure supplement 1.** SAP97 and SAP97/GluA1 interaction regulate GluA1 trafficking during chemical long term potentiation (cLTP).

**Figure supplement 1—source data 1.** Individual data values for the bar graphs in panels A and C.

insertion in rodents (***Lin et al., 2009***). These experiments were performed by directly visualizing individual insertion events of the AMPAR subunit GluA1 at the PM. It was thus particularly interesting to study if the 4.1 N/GluA1 interaction contributes to the regulation of GluA1 IT during synaptic plasticity or only to the exocytosis of GluA1. We next studied the contribution of 4.1 N to the cLTP-induced regulation of GluA1 IT. In basal conditions, when GluA1 is not associated with 4.1 N (GluA1-AA), IT

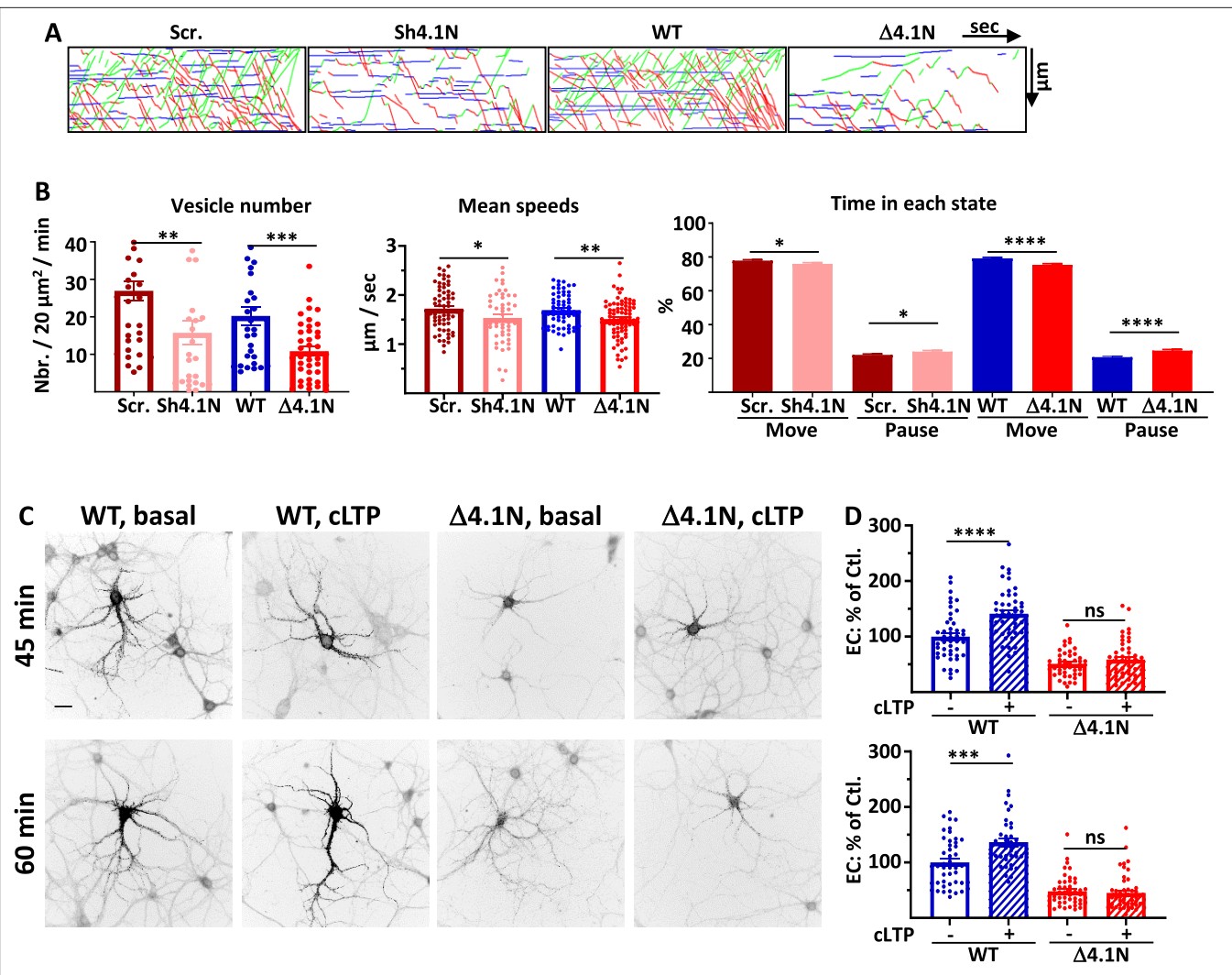

**Figure 5.** 4.1N and 4.1N/GluA1 interaction regulate GluA1 trafficking during chemical long term potentiation (cLTP). (**A**) Representative traced kymographs for ARIAD-TdTom-GluA1-WT co-transfected with scramble-RNA (Scr.) or with sh-4.1N (Sh4.1N), ARIAD-TdTom-GluA1-WT alone (WT) and ARIAD-TdTom-GluA1-Δ4.1N (Δ4.1N). (**B**) Intracellular transport (IT) parameters of GluA1 in the different conditions as indicated 25-40 min after induction of cLTP: Vesicle number (vesicles/20 µm²/min; scr.: 26.94 +/−2.59, Sh4.1N: 15.75 +/−3.13; WT: 20.25 +/−2.42, Δ4.1N: 10.84 +/−1.23), mean speeds (µm/s; scr.: 1.72 +/−0.05, Sh-4.1N: 1.53 +/−0.07, WT: 1.69 +/−0.04; Δ4.1N: 1.51 +/−0.04), and time in each state (Move and Pause) (Move: scr.: 77.81 +/−0.50%, Sh-4.1N: 75.95 +/−0.67%; pause: scr.: 22.18 +/−0.50%, Sh-4.1N: 24.02 +/−0.67%, and Move: WT: 79.23 +/−0.54%, Δ4.1N: 75.34 +/−0.67%; pause: WT: 20.76 +/−0.54%, Δ4.1N: 24.70 +/−0.67%) (n=3). (**C**) Representative images of neurons for each condition 45-60 min after the addition of the AL, 25-40 min after induction of cLTP. Scale bar: 25 µm. (**D**) Quantification of live immunolabeling before and after induction of cLTP for the GluA1-WT (WT) and GluA1-Δ4.1N (Δ4.1N) mutant (% PM localization at 45 min.: WT before LTP: 100 +/−6.51, after LTP: 141.06 +/−7.17, Δ4.1N before LTP: 50.80 +/−3.37, after LTP: 58.59 +/−5.14; % PM localization at 60 min: WT before LTP: 100 +/−6.52, after LTP: 136.52 +/−7.47, Δ4.1N before LTP: 47.89 +/−3.98, after LTP: 45.04 +/−4.52) (n=3). Control (Ctl.) corresponds to extracellular (EC) labeling of the WT before LTP normalized to 100%.

The online version of this article includes the following source data and figure supplement(s) for figure 5:

**Source data 1.** Individual data values for the bar graphs in panels B and D.

**Figure supplement 1.** 4.1 N and 4.1 N/GluA1 interaction regulate GluA1 trafficking during chemical long term potentiation (cLTP).

is normally compared to that of GluA1-WT. However, the interaction between GluA1 and 4.1 N is important for the exit of GluA1 from the ER/Golgi and the insertion at the PM. We thus analyzed the transport of GluA1 during cLTP upon downregulation of 4.1 N by sh-4.1 N and when the interaction between 4.1 N and GluA1 is abolished such as for the GluA1-Δ4.1N mutant. As for SAP97, we applied the cLTP protocol in cultured rat hippocampal neurons (*Hangen et al., 2018*) and compared IT of GluA1 or mutant 25–40 min after induction of cLTP (*Figure 5*, *Figure 5—figure supplement 1*).

After induction of cLTP, the number of vesicles was decreased compared to the corresponding controls either by decreasing the expression of 4.1 N or by inhibition of the binding by expression of Δ4.1N (*Figure 5A and B*, *Figure 5—figure supplement 1A*). Without 4.1 N, the decrease in the number of vesicles transporting GluA1 was comparable after cLTP and in basal condition. In the same way, inhibition of binding of 4.1 N on GluA1 decreased similarly IT of GluA1 in basal and cLTP conditions (% decrease of vesicles: sh-4.1N/scr. in basal: 45.6%, after cLTP: 41.5%; Δ4.1N/WT in basal: 43.4%, after cLTP: 46.4%). The mean speed of IT 25–40 min after induction of cLTP was impacted by the expression of the sh-4.1N (*Figure 5B*). It was also decreased when we expressed GluA1-Δ4.1N compared to the speed of GluA1-WT. This difference in speed translated into a decrease in the moving state to the benefit of pausing time (*Figure 5B*).

We then analyzed the exocytosis of newly synthesized GluA1 after induction of cLTP (*Figure 5C and D*). The induction of cLTP allows an increase of the externalization of GluA1 WT protein at 45 min and 60 min after the addition of AL like previously shown (*Hangen et al., 2018*). In the same conditions, cLTP had no impact on the externalization at the PM of GluA1-Δ4.1N.

Altogether, both suppressing SAP97 and 4.1 N interactions with GluA1 prevented the cLTP-induced regulation of GluA1 ER/Golgi exit, intracellular transport, and plasma membrane exit (*Supplementary file 1*).

We found that the parameters of GluA1-Δ4.1N IT are decreased during cLTP. During LTP, protein kinase C (PKC) phosphorylation of the serine 816 ($S_{816}$) and serine 818 ($S_{818}$) residues of GluA1 enhanced 4.1 N binding to GluA1 and facilitated GluA1 insertion at the PM (*Lin et al., 2009*). We thus decided to study the impact of cLTP when GluA1 cannot be phosphorylated at these two sites by using the AA mutant that does not bind 4.1 N (*Figure 6*, *Figure 6—figure supplement 1*). This construct allowed us to study directly the impact of the binding of 4.1 N after induction of cLTP together with the impact on exocytosis of GluA1 in both conditions.

On each set of experiments, we verified that the cLTP protocol increased the number of vesicles on a few cells expressing ARIAD-TdTom-GluA1 (*Hangen et al., 2018*; *Figure 6A*). After cLTP the number of vesicles for the AA mutant was decreased by 64.3% compared to the WT (*Figure 6B and C*, *Figure 6—figure supplement 1A*). The mean speed was significantly reduced for the AA mutant compared to the WT (*Figure 6C*, *Figure 6—figure supplement 1B*). This difference in speed has for consequence that the time spent in the moving state is decreased for the AA mutant to the benefit of the time in pause (*Figure 6C*). We analyzed the frequency distribution of speed and found that the pool of receptors going fast (2–4 μm/s) is decreased for the AA mutant to the benefit of the slow vesicles (*Figure 6D*). This is the case for both OUT and IN directions (*Figure 6—figure supplement 1C*). Thus, in cLTP condition the binding of 4.1 N on the C-ter. domain of GluA1 is necessary for the transport of high-velocity vesicles.

Finally, we analyzed the exocytosis of the neosynthesized GluA1 before and after induction of cLTP (*Figure 6E and F*). cLTP increases the exocytosis of GluA1-WT but does not change the localization at the PM of GluA1-AA. For this mutant the externalization of GluA1 is decreased both at basal state and inducing cLTP does not change the rate of externalization of the newly synthesized GluA1.

During basal transmission, the interaction between GluA1 and 4.1 N is only important for the exocytosis of the receptor since its vesicle number and mean speed of IT are not affected by the mutation. However, during cLTP the IT properties are largely decreased together with its exocytosis, showing a fundamental role of this interaction during cLTP (*Supplementary file 1*).

## Discussion

The characteristics of GluA1 IT are reproducible in basal conditions and heavily regulated during cLTP, presumably allowing for the tuning of newly synthesized AMPAR numbers at the PM (*Hangen et al., 2018*). In this study, we analyzed the impact of the expression and binding of 4.1 N and SAP97, specifically associated with the intracellular C-ter. domain of GluA1, on the regulation of AMPAR IT (*Figure 7*). We detected a differential role between 4.1 N or SAP97 binding on the GluA1 subunit of AMPAR. These interactions regulate different parameters of IT and exocytosis of neosynthesized GluA1 during basal transmission or during cLTP.

Independently decreasing expression of 4.1 N or SAP97 inhibited GluA1 IT as well as the exocytosis of the receptor both in basal transmission and after cLTP. For IT regulation, the impact of the reduction of 4.1 N was all less drastic than for SAP97. This was the case for the decrease in the number

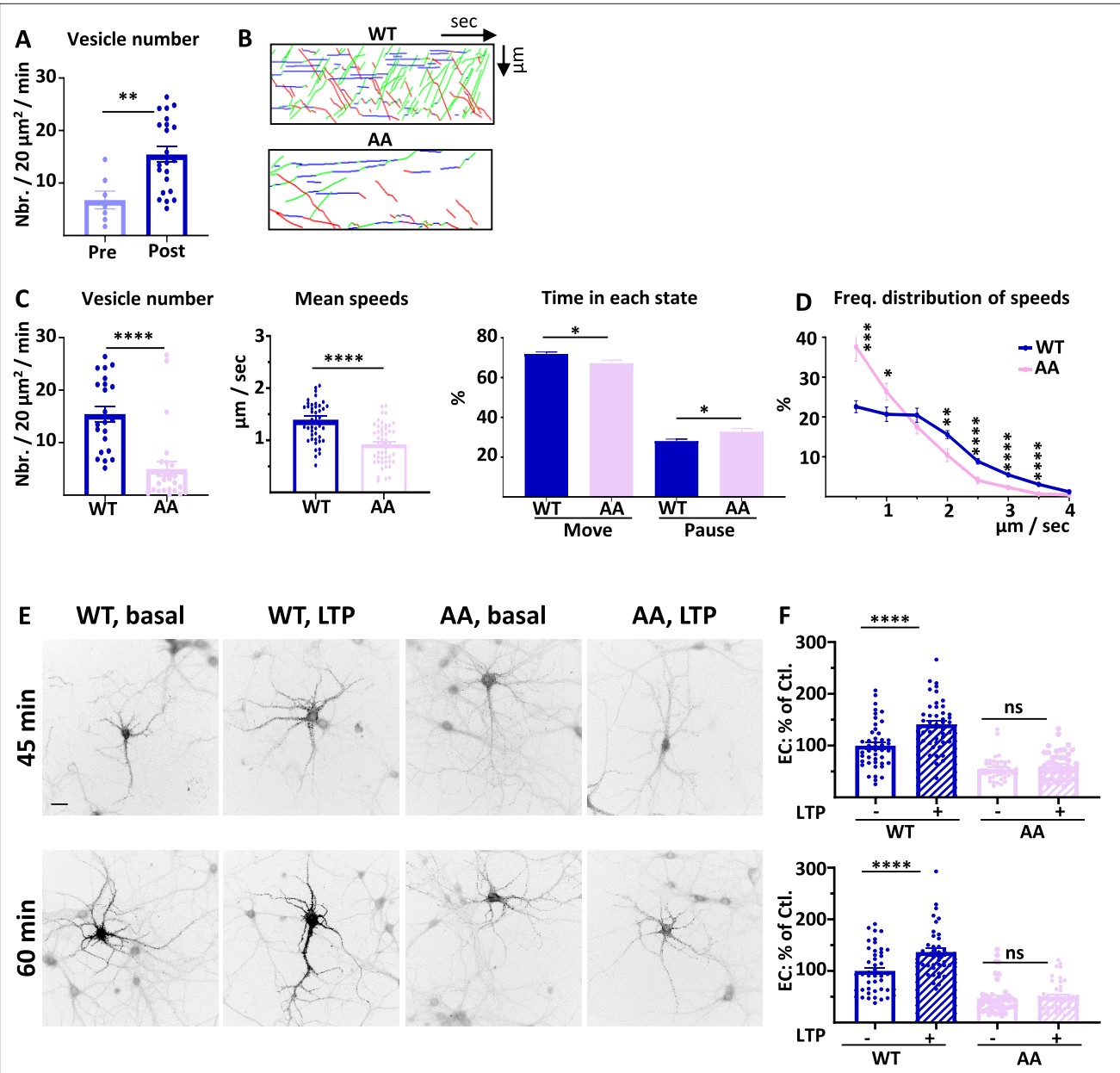

**Figure 6.** 4.1 N/GluA1 interaction drives intracellular transport of GluA1 during chemical long term potentiation (cLTP). (**A**) Number of vesicles before (Pre) and after (Post) induction of cLTP for ARIAD-TdTom-GluA1-WT (vesicles/20 μm²/min; Pre cLTP: 6.77 +/−1.70, Post cLTP: 13.99 +/−1.52). (**B**) Representative traced kymographs for ARIAD-TdTom-GluA1-WT (WT) and ARIAD-TdTom-GluA1-AA (AA) 25–40 min after induction of cLTP. (**C**) IT parameters of GluA1 in the different conditions as indicated 25–40 min after induction of cLTP: Vesicle number (vesicles/20 μm²/min; GluA1-WT: 13.99 +/−1.52, AA: 4.99 +/−1.38), mean speeds of the vesicles (μm / sec: WT: 1.37 +/−0.06, AA: 0.92 +/−0.05) and time in each state (Move and Pause) (Move: WT: 71.68 +/−0.97%, AA: 67.22 +/−1.57%; pause: WT: 28.08 +/−0.97%, AA: 32.78 +/−1.57%). (**D**) Frequency distribution of speed for the two proteins after induction of cLTP.( **E**) Representative images for each condition 45 and 60 min after the addition of the AL, 25 and 40 min. after induction of cLTP. Scale bar: 25 μm. (**F**) Quantification of live labeling of GluA1 in WT and AA conditions, before and after cLTP (% PM localization at 45 min: WT before LTP:100 +/−6.51, after LTP: 141.06 +/−7.17, AA before LTP: 54.84 +/−3.27, after LTP: 60.31 +/−3.70; % PM localization at 60 min: WT before LTP: 100 +/−6.52, after LTP: 136.52 +/−7.47, AA before LTP: 47.17 +/−4.64, after LTP: 50.17 +/−4.04).

The online version of this article includes the following source data and figure supplement(s) for figure 6:

**Source data 1.** Individual data values for the bar graphs in panels A, C, D and F.

**Figure supplement 1.** 4.1 N/GluA1 interaction drives intracellular transport of GluA1 during chemical long term potentiation (cLTP).

**Figure supplement 1—source data 1.** Individual data values for the bar graphs in panels B and C.

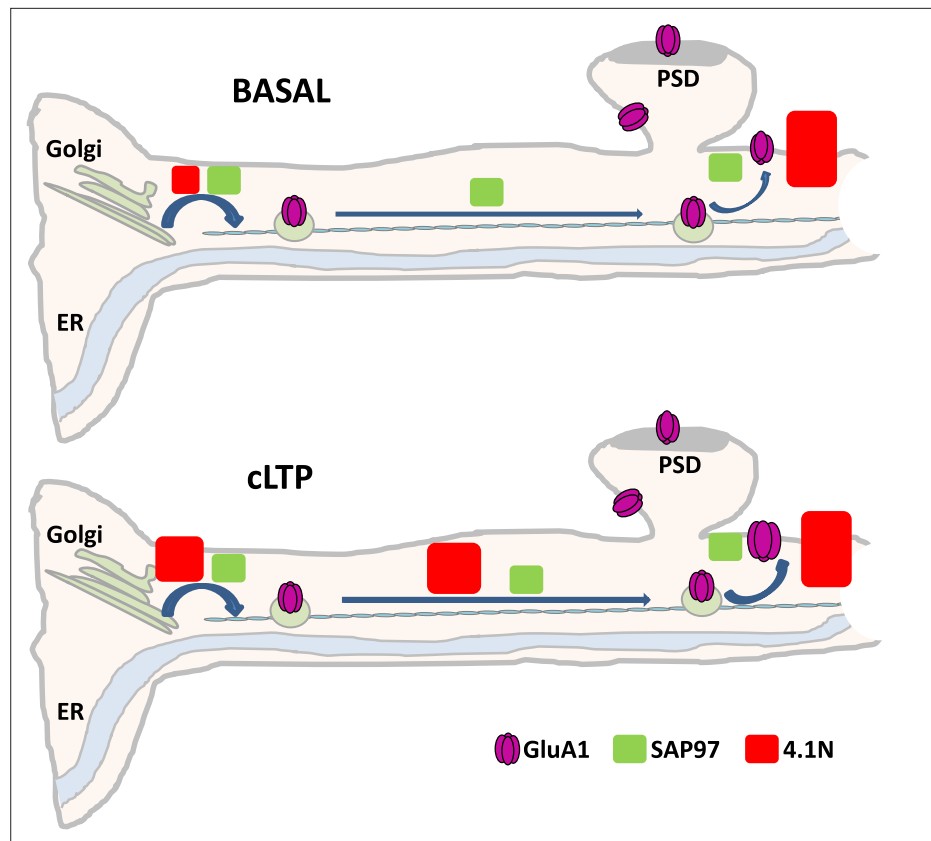

**Figure 7.** Diagram showing the differential roles of 4.1N/GluA1 and SAP97/GluA1 in basal transmission and during chemical long term potentiation (cLTP). In basal condition, upper panel, SAP97 participates in the different phases of intracellular transport (IT), and localization of GluA1 at the plasma membrane (PM) is decreased. The interaction between 4.1N and GluA1 is fundamental only for the exocytosis of the receptor at the PM. After induction of cLTP, down panel, SAP97 has the same effect than in the basal state. However, 4.1N regulates the exit from the ER/Golgi, the IT, and the exocytosis of GluA1.

of vesicles released upon the addition of AL as well as for the vesicle speed and for the increase in pausing time. On the contrary, we found that the externalization of GluA1 at the PM was equally inhibited by the absence of 4.1 N or SAP97.

The approach we are using allows to study the intracellular transport of neosynthesized GluA1. It however bears some shortcomings. For example, the fact that is an over-expression system and that vesicles are released from the ER synchronously is quite different from in vivo conditions and might affect trafficking. Our cDNAs are cloned in a Tet-ON system allowing us to express our proteins for a short time in order to avoid strong over-expression. In addition, we have previously shown that transport properties measured with the ARIAD release system release are indistinguishable from that of constitutively expressed receptors (*Hangen et al., 2018*). Another limitation of our current study is that we are studying the transport properties of expressed GluA1 that could form either homomers or associate with endogenous subunits, and most probably GluA2. Importantly, GluA1 homomers are involved in various forms of LTP. Moreover, GluA1 C-ter. domain is specifically associated with proteins implicated in AMPAR transport such as 4.1 N and SAP97.

AMPAR is part of a macromolecular complex composed of a broad range of AMPAR interacting proteins including transmembrane auxiliary proteins and cytosolic partners associated with the C-terminal domain of AMPAR subunits. Several interactors were shown to affect biogenesis, AMPAR trafficking, and channel properties. Most of these proteins revealed preferred binding to specific AMPAR subunits, like SAP97 on GluA1 and GRIP1/PICK1 on GluA2 and GluA3. In *C. elegans*, motor-mediated transport is the major mechanism for the delivery, removal, and redistribution of GLR-1 glutamate receptors (*Hoerndli et al., 2015*). The microtubule-dependent motor UNC-116 (homolog of mammalian KIF5) drives the delivery and the removal of AMPAR, while UNC-43 (homolog of mammalian

CaMKII) plays an essential role in modulating the transport of AMPAR between the cell body and the insertion or removal of synaptic AMPAR (*Hoerndli et al., 2015*). In vertebrates, AMPAR interacts with KIF5 through GRIP1 (*Setou et al., 2002*). AMPAR-mediated synaptic transmission depends on both dynein and kinesin superfamily proteins (*Kim and Lisman, 2001*) and an increase in AMPAR number at the PM during LTP depends also on their secretory transport (*Esteves da Silva et al., 2015*). Other motors have also been implicated in AMPAR transport, such as Myosin-VI, an actin-dependent motor protein (*Wu et al., 2002*). SAP97 may serve as a molecular link between GluA1 and myosin-VI during the dynamic translocation of AMPAR to and from the post-synapse in an activity-dependent manner (*Wu et al., 2002*; *Nash et al., 2010*).

Whether the C-ter. domain of GluA1 plays any role in synaptic regulation is still uncertain. C-ter. domains of GluA1 and GluA2 are necessary and sufficient to drive NMDA receptor-dependent LTP and LTD, respectively (*Zhou et al., 2018*). Using a single-cell molecular replacement approach and long-expression time, the authors found no requirement for the GluA1 C-ter. domain of GluA1 for basal synaptic transmission nor for LTP (*Granger et al., 2013*). Other work observed a substantially reduced insertion frequency after expression of SEP-GluA1 and TIRF microscopy when the C-ter. domain was deleted (*Lin et al., 2009*). In our model, we found that deleting the C-ter. domain of GluA1 inhibits almost completely IT and PM localization of newly synthesized GluA1. However, we could still detect rare vesicles transporting the subunit. Our experiments were performed until 120 min after the addition of the ligand. We cannot rule out that after 24–48 hr some AMPAR could reach the synapse. For the measurement of surface expression of the $\Delta 7$ and $\Delta 4.1N$ mutants, all the values are expressed as a percentage of the WT expression level analyzed in the same experiments. We cannot, however, completely rule out that an increased endocytosis of each mutant is responsible for the reduced surface expression instead of a direct impact on exocytosis.

To analyze the impact of the interaction between GluA1 and 4.1 N, we used two mutants of GluA1. The first one is a complete deletion of the 4.1 N binding site, corresponding to a deletion of 14 amino acids close to the last transmembrane domain of GluA1. With this mutant, during basal transmission, we found a decrease in the number of vesicles transporting GluA1, a normal IT, and a decrease in exocytosis of GluA1. These effects could be due to the different lengths of the C-ter. domain of GluA1 or to the unbinding of 4.1 N with GluA1. In order to keep the same size as the C-ter. domain, we performed IT and exocytosis experiments with the GluA1-$S_{816}$A-$S_{818}$A (AA) mutant that does not bind 4.1 N. With this mutant, during basal transmission, only the exocytosis of the protein is impacted. This shows that the interaction between GluA1 and 4.1 N is only necessary for receptor exocytosis, without affecting its IT during basal transmission. The interaction with SAP97 has already a role during the transport of GluA1. In this case, externalization of GluA1 is impacted either because of the reduced vesicle number or because of a direct effect on exocytosis. After cLTP, we observed a decrease in IT properties when the interaction of 4.1 N/GluA1 was abolished. The effect of the SAP97/GluA1 interaction during cLTP stays the same as in basal conditions.

Previous studies have provided conflicting data regarding SAP97's influence on synaptic function. Mutant mice expressing GluA1-$\Delta 7$ were found to have normal glutamatergic neurotransmission in hippocampal CA1 pyramidal neurons (*Zhou et al., 2008*). In another study, SAP97 isoforms appeared to regulate the ability of synapses to undergo plasticity by controlling the surface distribution of AMPA and NMDA receptors (*Li et al., 2011*). SAP97 directs GluA1 forward trafficking from the Golgi network to the PM. Myosin VI and SAP97 are thought to form a trimeric complex with GluA1, with SAP97 acting as an adaptor between GluA1 and myosin VI to transport AMPA receptors to the post-synaptic plasma membrane (*Wu et al., 2002*). SAP97 only interacts with GluA1 early in the secretory pathway during its forward trafficking to the PM, suggesting that SAP97 acts on GluA1 solely before its synaptic insertion and that it does not play a major role in anchoring AMPAR at synapses (*Sans et al., 2001*). SAP97 has also been shown to be associated with motor proteins such as KIF1B (*Mok et al., 2002*). Here, we found that SAP97 has a major role in the IT of AMPAR. Inhibiting the expression of SAP97 decreases IT and externalization of neosynthesized GluA1. This effect remains after the induction of cLTP. This result is in accordance with the fact that SAP97 has been already shown to be associated with AMPAR during its biogenesis (*Sans et al., 2001*).

4.1 N may function to confer stability and plasticity to the neuronal membrane via interactions with multiple binding partners, including the spectrin-actin-based cytoskeleton, integral membrane channels, and receptors. Phosphorylations at both $S_{816}$ and $S_{818}$ residues regulate activity-dependent

GluA1 insertion, by affecting the interaction between 4.1 N and GluA1. In hippocampal neurons, when 4.1 N does not bind GluA1 (expression of GluA1-$S_{816}$A-$S_{818}$A), a substantially lower insertion frequency is detected by TIRF microscopy. During LTP these serines are phosphorylated, thus increasing the binding of 4.1 N to GluA1 and exocytosis of GluA1. 4.1 N is an important player in the expression of LTP, but doesn't affect basal synaptic transmission (*Lin et al., 2009*). With our experiments, we show that in basal conditions only the exocytosis of neosynthesized GluA1 is inhibited when it does not bind to 4.1 N without affecting its IT. However, during cLTP exocytosis and IT properties of GluA1 are dependent of the binding of 4.1 N on GluA1.

We show differential roles of 4.1 N/GluA1 and SAP97/GluA1 in basal transmission and during cLTP. In basal condition (upper panel *Figure 7*) binding of SAP97 to GluA1 participates in the exit from the ER/Golgi and to intracellular transport of GluA1. By a consequence, the number of GluA1 at the PM is decreased (only by 30% compared to the control condition). The interaction between 4.1 N and GluA1 is necessary only for the exocytosis of the receptor at the plasma membrane (60% decrease compared to the control) having no effect on IT. After induction of cLTP (bottom panel *Figure 7*) SAP97 has the same effect as in the basal state. However, after cLTP the role of the interaction between 4.1 N and GluA1 becomes crucial for all phases from the ER to the externalization of AMPAR. By analyzing the GluA1-$S_{816}$A-$S_{818}$A mutant we showed that 4.1 N/GluA1 interaction regulates the exit of the receptor from the ER/Golgi, the intracellular transport, and the exocytosis.

In this study, we uncover a new mechanism of regulation of AMPAR trafficking during basal synaptic transmission and synaptic plasticity. It will be important to decipher if this mechanism still all thru in vivo and whether its abnormal regulation is involved in some synaptic disruption.

## Materials and methods

### Molecular biology

cDNAs of interest are cloned in the ARIAD system (*Hangen et al., 2018*) and then in the Tet-on vector for cLTP experiments. Mutations and deletions are performed by directed mutagenesis and controlled by sequencing before use. sh-RNA against 4.1 N is a gift from Huganir's lab (*Lin et al., 2009*). Sh-RNA against SAP97 (sequence: GATATCCAGGAGCATAAAT) was cloned in an FHUG vector expressing also GFP.

### Cell culture

Sprague–Dawley pregnant rats (Janvier Labs) were killed according to the European Directive rules (2010/63/EU). Dissociated hippocampal neurons from E18 embryos were prepared as described previously (*Kaech and Banker, 2006*) at a density of 300,000 cells per 60 mm dish on poly-L-lysine pre-coated coverslips. Neurons were transfected with the cDNA using $Ca^{2+}$ method at 8–11 days in vitro. Proteins of interest are expressed for 3–6 days depending on the experiments. For cLTP experiments expression is started two days before the experiments by the addition of 500 nM of doxycycline in the media. All experiments were performed in accordance with the European guidelines for the care and use of laboratory animals, and the guidelines issued by the University of Bordeaux animal experimental committee (CE50; Animal facilities authorizations A3306940, A33063941).

COS-7 cells were maintained in Dulbecco's Modified Eagle's Medium (DMEM 4,5 G/L+GLUT & PYRUVATE 500, Eurobio) with 10% fetal bovine serum (Eurobio) and 1% L-glutamine (Gibco). Transfection was done with the Xtreme gene HP DNA transfection kit (Roche) following the manufacturer's protocol.

### Immunoprecipitation

All subsequent steps were performed at 4 °C. Cells are solubilized in a lysis buffer all (in mM: 50 Hepes pH7.3, 0.5 EDTA, 4 EGTA, 150 NaCl, 1% Triton-X100 and 10 µg/mL antiproteases: Leupeptin, Pepstatin, Aprotinin, Pefabloc, Mg132) and centrifuged for 15 min at 15,000 G. Supernatant is cleared on the resin for 1 hr and incubated with the antibodies of interest for 2 hr followed by incubation with protein-A Sepharose overnight. Beads are rinsed with lysis buffer and a lysis buffer containing 500 mMol of NaCl. Beads are eluted with Laemmli sample buffer. Western blots are revealed with an Odyssey CLX machine (LI-Cor) by fluorescence method.

## Immunocytochemistry

After the addition of 1 µMol of D/D Solubilizer (ARIAD Ligand, AL) in the cell culture medium, neurons were incubated at a different time. Ten minutes before fixation extracellular labeling of tagged receptors was performed in live with a monoclonal anti-GFP (1/1000) or polyclonal anti-DSRed (1/1000) antibodies incubated in the media at 37 °C. Neurons are then fixed with 4% paraformaldehyde, 4% sucrose, rinsed with 50 mM of $NH_4Cl$, and transferred in PBS/BSA (1%). Secondary antibody anti-mouse Alexa-fluo588 or anti-rabbit Alexa-fluo488 was applied for 20 min at room temperature. Coverslips were mounted on glass slides in Fluoromount-G media (Lonza, ref: 0100–01). Images were collected on an upright Leica DM5000 epifluorescence microscopy with an LED light source using a 40 x oil objective.

## Quantification

For quantification of extracellular labeling of GFP-GluA1, surfaces of interest were drawn by hand following the total GFP labeling. Images were processed using ImageJ software (Rasband, W.S., ImageJ, U. S. NIH, Bethesda, Maryland, USA, http://imagej.nih.gov/ij/, 1997–2012). Briefly, quantifications were performed by first asking the user to draw sets of regions of interest: one in the background (devoid of any structure), one in the proximal, and one in the distal dendritic shaft. For each of them, average intensity and area were retrieved. Results are expressed as the mean intensity fluorescence at the PM. Statistics were performed on Prism software using an unpaired $t$-test. Value are in mean +/−SEM. Asterisks notify the following significance levels: $p < 0.05$ (*), $p < 0.01$ (**), $p < 0.001$ (***), and $p < 0.0001$ (****).

## Videomicroscopy

The videos were acquired on an inverted Leica DMI6000B, equipped with a spinning-disk confocal system (Yokogawa CSU-X1, beam lines: 491 nm, 561 nm), EMCCD camera (Photometrics Quantem 512), and a HCX PL Apo 100X1.4 NA oil immersion objective. The microscope was driven by the Metamorph software (Molecular Devices, Sunnyvale, USA) and the acquisition took place at 37 °C using a Life Imaging Services chamber. The coverslips were mounted in a Ludin chamber filled with 1 ml of tyrode media of appropriate osmolarity (15 mM glucose, 100 mM NaCl, 5 mM KCl, 2 mM $MgCl_2$, 2 mM CaCl2, 10 mM HEPES, 247 mOsm/l) onto an inverted microscope. 1 µM of AL was added in the tyrode to release the constructs from the ER.

For the basal experiments, during the first 30 min, the positions of the expressing cells were saved under the 40 X objective. Videos were acquired as follows in the red channel with the 100 X objective during the next 30 min: during the first 10 s, 10 images of 100 ms exposure were acquired, followed by the photobleaching of a portion of a dendritic shaft (~60 µm², 561 nm laser) followed by a 1 min stream of 600 frames of 100 msec exposure to image the transport of the vesicles. For the sh experiments an image was acquired in the green channel prior to the video to confirm the co-transfection of the cell. For each coverslip, 4–10 cells were imaged between 30–60 min of incubation with the ligand.

For cLTP experiments, the positions were saved during the first 20 min in a tyrode media (25 mM Glucose, 20 mM HEPES, 150 mM NaCl, 3.5 mM KCl, 2 mM MgCl2, 2 mM CaCl2, 304 mosm/l). The tyrode was then replaced by a magnesium-free tyrode (25 mM glucose, 20 mM HEPES, 150 mM NaCl, 3.5 mM KCl, 2mMCaCl2, 200 µM glycine, 20 µm bicuculline, 300 mosm/l) for 4 min to induce cLTP. Then the original tyrode was put back in the Ludin. The videos were acquired 25 min after cLTP induction, using the same acquisition parameters as for basal. Cells having less than five vesicles/20 µm²/min. were excluded from the IT analysis of the GluA1 WT condition.

Analysis of the videos is performed using ImageJ program with a help of 'Kymo-Tool-Set' software homemade (*Hangen et al., 2018*) that can be downloaded, together with its source code from https://github.com/fabricecordelieres/IJ-Plugin_KymoToolBox (*Cordelières, 2020*). This plugin allows to trace kymographs (distance versus time) of each vesicle and allows to have the parameters of each vesicle such as the number of vesicles, speed of each vesicle, and time spends in each direction (OUT and IN). Statistics were performed on Prism software using an unpaired *t*test. Values are in mean +/−SEM. Asterisks notify the following significance levels: $p < 0.05$ (*), $p < 0.01$ (**), $p < 0.001$ (***), and $p < 0.0001$ (****).

## Materials and antibodies

ARIAD ligand (AL), D/D Solubilizer from Takara, ref: 635054.

Anti-GFP from Sigma Aldrich (ref: 11814460001), anti-Ds-Red from Ozyme (ref: 632496), anti-myc from Merck Millipore (ref: 6549), anti-4.1N from BD Biosciences (ref: 611836) anti-SAP97 from NeuroMab (ref:75–030), anti-GluA1 from NeuroMab (ref: 75–327). Secondary antibodies from Molecular Probes: goat anti-mouse Alexa Fluor-568 (ref: A11004), goat anti-rabbit Alexa Fluor-488 (ref: A11008); from Li-Cor: goat anti-mouse (ref: 926–32210), goat anti-rabbit (ref: 926–68021).

## Acknowledgements

The microscopy was done in the Bordeaux Imaging Center (BIC), a service unit of the CNRS-INSERM and Bordeaux University, a member of the national infrastructure 'France BioImaging.' We specifically thank the help of Magalie Mondin and Fabrice Cordelières. We like to thank the IINS Cell Biology facility, especially Emeline Verdier, Christelle Breillat, Sophie Daburon and Nicolas Chevrier, for cell culture and plasmid production. This work is currently supported by funding from the Ministère de l'Enseignement Supérieur et de la Recherche, Centre National de la Recherche Scientifique, ERC Grant #787340 Dyn-Syn-Mem, and the Conseil Régional de Nouvelle Aquitaine.

## Additional information

### Funding

| Funder | Grant reference number | Author |
| --- | --- | --- |
| European Research Council | 787340 | Daniel Choquet |
| Ministère de l'Education Nationale, de l'Enseignement Superieur et de la Recherche | | Françoise Coussen |

The funders had no role in study design, data collection and interpretation, or the decision to submit the work for publication.

### Author contributions

Caroline Bonnet, Data curation, Software, Formal analysis, Validation, Investigation, Methodology; Justine Charpentier, Natacha Retailleau, Data curation, Methodology; Daniel Choquet, Conceptualization, Formal analysis, Investigation, Writing – original draft, Project administration, Writing – review and editing; Françoise Coussen, Conceptualization, Formal analysis, Supervision, Funding acquisition, Investigation, Methodology, Writing – original draft, Project administration, Writing – review and editing

### Author ORCIDs

Daniel Choquet ⓘ http://orcid.org/0000-0003-4726-9763
Françoise Coussen ⓘ http://orcid.org/0000-0002-3194-3058

### Decision letter and Author response

Decision letter https://doi.org/10.7554/eLife.85609.sa1
Author response https://doi.org/10.7554/eLife.85609.sa2

## Additional files

### Supplementary files

• Supplementary file 1. Comparisons vesicle number, mean speed, and externalization of the mutants. Values are calculated as a percentage of the corresponding WT GluA1 in the same set of experiments.

• Supplementary file 2. Statistical analysis of the results.

• MDAR checklist

## Data availability

All data needed to evaluate the conclusions in the paper are present in the paper and/or the supplementary materials. The data generated, analyzed, and used for this study can be accessed at https://doi.org/10.5281/zenodo.7780877.

The following dataset was generated:

| Author(s) | Year | Dataset title | Dataset URL | Database and Identifier |
|---|---|---|---|---|
| Coussen F | 2023 | Regulation of different phases of AMPA receptor intracellular transport by 4.1N and SAP97 | https://doi.org/10.5281/zenodo.7780877 | Zenodo, 10.5281/zenodo.7780877 |

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
