## [Editor Report]

This important study by Bonnet et al. addresses the question of how AMPA receptor numbers at the synapse are regulated during basal conditions and during chemically induced Long Term Potentiation. Specifically, the study aims to determine the molecular mechanisms that contribute to the intracellular trafficking of AMPA receptors and determine their insertion into the synaptic plasma membrane. Using compelling methodology, the authors dissect the distinct roles of two proteins that bind to the C-terminal domain of the AMPA receptor subunit GluA1: 4.1N and SAP97. The findings will be of interest to anyone trying to understand molecular events contributing to synaptic plasticity in health and disease, and more broadly, the method could be adapted for tracking intracellular movements of a wide range of proteins.

---

## [Decision Letter]

**Decision letter after peer review:**

Thank you for submitting your article "Regulation of different phases of AMPA receptor intracellular transport by 4.1N and SAP97" for consideration by *eLife*. Your article has been reviewed by 3 peer reviewers, including Lejla Zubcevic as the Reviewing Editor and Reviewer #1, and the evaluation has been overseen by Richard Aldrich as the Senior Editor.

Essential revisions:

1) Since the approach used here could be used for a variety of proteins and is hence of interest to the wider scientific community, I think it would be very important to clearly discuss the shortcomings of the approach. For example, the fact this is an over-expression system and that vesicles are released from the ER synchronously is quite different from in vivo conditions and might affect trafficking.

2) In terms of AMPARs, it should be clearly noted that the subunit composition used here is not very represented in vivo. In fact, hippocampal AMPARs are mainly heteromers composed of A1/A2, A3/A2, or even A1/A2/A3 subunits (Yu, Nature, 202, https://doi.org/10.1038/s41586-021-03540-0), which is very likely to affect their trafficking properties.

3) The number of released vesicles from the ER and externalisation of AMPARs are considered separately throughout the manuscript and conclusions are made about whether an interaction affects externalisation or intracellular trafficking based on this. This works really well in the case of double Ala mutant (S816A S818A) of AMPARs, where the number of released vesicles is unchanged compared to the non-mutated AMPARs, but there is less externalisation in basal conditions. However, I was wondering whether a reduced number of released vesicles could lead to less externalisation simply because there are fewer vesicles to deliver to the plasma membrane and not because something is wrong with the externalisation process per se.

4) In experiments where surface localisation is measured over extended periods of time (e.g. over 2 hours in Figure 2D), how sure can one be there is no endocytosis happening during this time and that this not contributing to the decrease in surface intensity observed in some AMPAR mutants compared to the non-mutated AMPARs?

*Reviewer #1 (Recommendations for the authors):*

I have the following suggestions for the authors.

1. Data presentation:

1.1. A table (or similar) to summarize and directly compare the results of the various conditions, and types of measurements might be helpful.

1.2. Axis labels in "% prot/scr" and "%EC/scr" graphs – perhaps rename to something that is easier to understand without having to refer to the figure legends? It'd make the figures easier to read if the graphs contained all the information required to interpret the data.

1.3. Many sentences are punctuated with long parentheticals, for example, "(% decrease of vesicles: control/Sh-SAP97 in basal: 78.4 %, after cLTP: 67.1%; control/∆7 in basal: 27.2%, after cLTP: 33.8 %)." This interrupts the thought while reading. Perhaps referring to a figure panel instead would be more effective.

1.4. Several typos etc. present in the text – a re-reading with a fresh pair of eyes would probably deal with most of this.

2. Methods could use some more details. For example, how were the vesicle speeds calculated?

3. There is only a paper reference for the homemade software for kymograph analysis- a note should be made about where this software can be obtained (like a public repository? Or by reasonable request?) Please see these notes:

https://reviewer.elifesciences.org/author-guide/journal-policies

Software

Authors are required to follow the guidelines developed by PLOS if new software or a new algorithm is central to the submission; for example, authors must confirm that software conforms to the Open Source Definition and is deposited in an appropriate public repository. To ensure that software can be reproduced without restrictions and that authors are properly acknowledged for their work, authors should license their code using an open-source license.

Authors are encouraged to use version control services such as GitHub, GitLab, and SourceForge. *eLife* archives code accompanying *eLife* publications which has been deposited on GitHub or another version control service at Software Heritage. Binary files ("non-text files", such as images, zip files, or program data) should be kept to a minimum and, if possible, they should not exceed 50MB. Please try to avoid files larger than 100MB as they will require special handling.

Software should be included as a reference and cited in the article.

*Reviewer #3 (Recommendations for the authors):*

The authors do not make it clear the level of over-expression of GluA1 and whether they are only expressing GluA1 alone or in combination with GluA2.

In figures looking at the surface expression of GluA1 the morphology of the neurons appears to be impacted, with decreased cell size and dendritic arborization, most likely because of the low expression of surface expression GluA1. Do all of these experiments contain GFP cell fills to better show the cell and dendritic morphology

This sentence on page 9 seems to be incorrect or confusing "The extracellular labeling of GluA1 is less important when the site of interaction of 4.1N is deleted than when the site of interaction of SAP97 is deleted." It should be corrected or clarified.

Why do they represent surface expression as mean intensity or %EC/WT in different figures/panels? The presentation of the data should be consistent.

---

## [Author Response]

Essential revisions:1) Since the approach used here could be used for a variety of proteins and is hence of interest to the wider scientific community, I think it would be very important to clearly discuss the shortcomings of the approach. For example, the fact this is an over-expression system and that vesicles are released from the ER synchronously is quite different from in vivo conditions and might affect trafficking.

We have added a paragraph in the discussion to discuss these points.

2) In terms of AMPARs, it should be clearly noted that the subunit composition used here is not very represented in vivo. In fact, hippocampal AMPARs are mainly heteromers composed of A1/A2, A3/A2, or even A1/A2/A3 subunits (Yu, Nature, 202, https://doi.org/10.1038/s41586-021-03540-0), which is very likely to affect their trafficking properties.

We have added a paragraph in the discussion to discuss these points.

3) The number of released vesicles from the ER and externalisation of AMPARs are considered separately throughout the manuscript and conclusions are made about whether an interaction affects externalisation or intracellular trafficking based on this. This works really well in the case of double Ala mutant (S816A S818A) of AMPARs, where the number of released vesicles is unchanged compared to the non-mutated AMPARs, but there is less externalisation in basal conditions. However, I was wondering whether a reduced number of released vesicles could lead to less externalisation simply because there are fewer vesicles to deliver to the plasma membrane and not because something is wrong with the externalisation process per se.

We have added a sentence in the fourth paragraph of the discussion.

4) In experiments where surface localisation is measured over extended periods of time (e.g. over 2 hours in Figure 2D), how sure can one be there is no endocytosis happening during this time and that this not contributing to the decrease in surface intensity observed in some AMPAR mutants compared to the non-mutated AMPARs?

All the values are expressed as a percentage of the WT expression level analyzed in the same experiments. We cannot however completely rule out that an increased endocytosis of each mutants is responsible for the reduced surface expression. A sentence has been added in the discussion on this point.

Reviewer #1 (Recommendations for the authors):I have the following suggestions for the authors.1. Data presentation:1.1. A table (or similar) to summarize and directly compare the results of the various conditions, and types of measurements might be helpful.

We thank very much the reviewer for this suggestion. We have added a summary table (Supplemental Table 1) in order to compare the different mutants. We have cited this table in the different Results sections.

1.2. Axis labels in "% prot/scr" and "%EC/scr" graphs – perhaps rename to something that is easier to understand without having to refer to the figure legends? It'd make the figures easier to read if the graphs contained all the information required to interpret the data.

We have changed the axis levels and explain in the legends the corresponding labels.

1.3. Many sentences are punctuated with long parentheticals, for example, "(% decrease of vesicles: control/Sh-SAP97 in basal: 78.4 %, after cLTP: 67.1%; control/∆7 in basal: 27.2%, after cLTP: 33.8 %)." This interrupts the thought while reading. Perhaps referring to a figure panel instead would be more effective.

We thank very much the reviewer for this suggestion. We have inserted all the values in the text of the figure legends. Indeed, this change makes the reading of the results clearer.

1.4. Several typos etc. present in the text – a re-reading with a fresh pair of eyes would probably deal with most of this.

We have corrected typos in the text.

2. Methods could use some more details. For example, how were the vesicle speeds calculated?

We gave more details on the analysis and quantification of the intracellular transport in the Method section.

3. There is only a paper reference for the homemade software for kymograph analysis- a note should be made about where this software can be obtained (like a public repository? Or by reasonable request?)

We have added the reference of the Kymo-Tool-Set we used for the analysis and quantification of the intracellular transport vesicles. We have added a link to the source code to have access to the plugin.

Reviewer #3 (Recommendations for the authors):The authors do not make it clear the level of over-expression of GluA1 and whether they are only expressing GluA1 alone or in combination with GluA2.

We have added a paragraph in the discussion to discuss these points.

In figures looking at the surface expression of GluA1 the morphology of the neurons appears to be impacted, with decreased cell size and dendritic arborization, most likely because of the low expression of surface expression GluA1. Do all of these experiments contain GFP cell fills to better show the cell and dendritic morphology

The expressed DNA codes for a transmembrane span-FKB aggregation domains-Furin site-GFP-GluAs. After addition of the ligand live immunolabeling is performed with an antiGFP and a red secondary antibody. The green color corresponds to the expressed proteins in the ER, Golgi and vesicles and not to soluble GFP to fill the cells. It is thus impossible to see and quantify the morphology of the neurons expressing different mutant of GluA1. We apologize for that.

This sentence on page 9 seems to be incorrect or confusing "The extracellular labeling of GluA1 is less important when the site of interaction of 4.1N is deleted than when the site of interaction of SAP97 is deleted." It should be corrected or clarified.

Indeed, our sentences were not very clear, we thank the reviewer for this remark. We have changed these three sentences.

Why do they represent surface expression as mean intensity or %EC/WT in different figures/panels? The presentation of the data should be consistent.

We represented the first immunocytochemistry with a graph in mean intensity. This graph allows us to show the curve of exit of GluA1 WT with a maximum exit between 30 and 60 min. the period during which we performed the experiments of transport. Then all our calculations and statistics are performed taking the exocytosis of the WT as 100% for each points. We then chose to represent the exit of the mutants compared directly to the WT. We have changed the labeling of the X axis by EC: % of WT or % of Ctl.